# Estimating and Explaining Model Performance When Both Covariates and Labels Shift

**Lingjiao Chen[1], Matei Zaharia[1], James Zou[1,2]**
[1]Department of Computer Sciences, [2]Department of Biomedical Data Science
Stanford University

## Abstract

Deployed machine learning (ML) models often encounter new user data that differs from their training data. Therefore, estimating how well a given model might perform on the new data is an important step toward reliable ML applications. This is very challenging, however, as the data distribution can change in flexible ways, and we may not have any labels on the new data, which is often the case in monitoring settings. In this paper, we propose a new distribution shift model, Sparse Joint Shift (SJS), which considers the joint shift of both labels and a few features. This unifies and generalizes several existing shift models including label shift and sparse covariate shift, where only marginal feature or label distribution shifts are considered. We describe mathematical conditions under which SJS is identifiable. We further propose SEES, an algorithmic framework to characterize the distribution shift under SJS and to estimate a model's performance on new data without any labels. We conduct extensive experiments on several real-world datasets with various ML models. Across different datasets and distribution shifts, SEES achieves significant (up to an order of magnitude) shift estimation error improvements over existing approaches.

## 1 Introduction

Deployed machine learning (ML) models often face new data different from their training data. For example, mismatch of deployment-development data in geographical locations [21], demographic features [16], and label balance [20] is widely observed and known to affect model performance. Thus, estimating and explaining how a model's performance changes on the new data is an important step toward reliable ML applications.

Estimating and explaining performance shift is challenging for several reasons, however. One major challenge is that the data distribution might shift in flexible ways. Another obstacle is that we often do not have labels on the new data, especially in ML monitoring applications. Without any assumption on the distribution shift, it's impossible to estimate how well the model would perform on the unlabeled new data. Previous work often assumes (i) label shift [22], where feature distributions conditional on the labels are fixed, or (ii) covariate shift [32], where label distributions conditional on features stay the same. However, we often do not know whether the real data shift is limited to label or covariate shift, and naively applying estimation methods designed for one shift may produce inaccurate assessments [26]. Moreover, labels and features may shift simultaneously in practice, invalidating these common assumptions.

To tackle the above challenges, here we propose a new distribution shift model, Sparse Joint Shift (SJS), to consider the joint shift of both labels and a few features. SJS assumes labels and a few features shift, but the remaining features' distribution conditional on the shifted features and labels is fixed. This unifies and generalizes sparse covariate shift and label shift: both of them are SJS, but some SJS is not label or sparse covariate shift (Figure 1). We describe mathematical conditions

36th Conference on Neural Information Processing Systems (NeurIPS 2022).

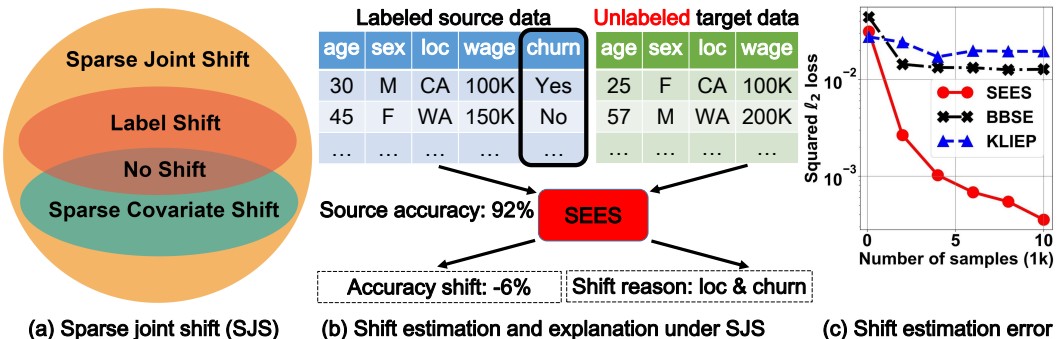

**Labeled source data**    **Unlabeled target data**

(a) Sparse joint shift (SJS)     (b) Shift estimation and explanation under SJS     (c) Shift estimation error

Figure 1: Overview of sparse joint shift (SJS). (a) Both label shift and sparse covariate shift are SJS, but SJS contains additional shifts as well. (b) illustrates SEES, a framework for performance shift estimation and explanation under SJS. Given labeled source and unlabeled target data, SEES exploits the joint shift modeled by SJS to estimate the model performance change and explain which factors drive the shift. In this example, the goal is to predict *churn*. (c) SEES significantly reduces the shift estimation error over existing methods when both labels and covariates shift.

under which SJS is provably identifiable: if the non-shifted features are weakly correlated, then the marginal feature distribution uniquely determines the joint distribution under SJS. This makes it possible to quantify the shift and estimate model performance on new data without any labels.

**When SJS occurs: a motivating example.**  Consider Alice, a data scientist who built an ML model for customer churn prediction. Two years later, churn rate rose in some states but dropped in others, while the distribution of other features given label (churn) and location (state) remained. This shift in customer distribution is a natural SJS and challenging to estimate without labels on the new data.

Furthermore, we propose SEES, an algorithmic framework for performance shift estimation and explanation under SJS. SEES exploits correlation shifts between features and labels modeled by SJS to improve performance estimation accuracy over existing methods, such as BBSE [22] for label shift and KLIEP [32] for covariate shift (Figure 1). It uses the identified shifted features and labels as a natural explanation of the performance shift. We present an extensive empirical evaluation on real world datasets with both simulated SJS and real shifts. Our experiments validate the effectiveness of SEES: the estimation error of SEES is often an order of magnitude smaller than that of previous state-of-the-art methods. All together, **our contributions are**:

1. We formulate and study sparse joint shift (SJS), a new distribution shift model that considers the joint shift of both labels and features. We show how it unifies and generalizes existing shift models and when it is identifiable with unlabeled data.

2. We propose SEES, a general framework for performance shift estimation and explanation under SJS. We design efficient substantiations of SEES for different data types. We also release our code[1].

3. We provide comprehensive empirical analysis of our methods. On real world datasets with natural distribution shifts, we found SEES leads to up to 66% performance estimation error reduction over standard approaches.

**Related Work.**  **Label shift:** Label shift has been observed and studied in various domains, such as epidemiologic [5], economics [23], and data mining [13]. Recently, there is an increasing interest in model evaluation and development under under label shift. For example, [22, 3] study how to quantify label shift and evaluate an ML model's performance under label shift accordingly. [40] gives an algorithm for active learning under label shift. Online adaptation to label shift [37] is also shown to be feasible. Label shift can be viewed as a subset of our proposed shift model SJS. Although simpler and easier to estimate, label shift does not capture the joint shift of labels and features. Thus, methods developed for label shift may not work well under SJS.

---

[1]`https://github.com/stanford-futuredata/SparseJointShift`

**Covariate shift:** Covariate shift [30, 38, 24] is perhaps the most widely adopted assumption in data distribution shift. Since studied in the seminal work [30], various methods have been proposed to estimate covariate shift, including KLIEP [33], KMM [15], and IWCV [31]. Adaptation to covariate shift has been found useful in many applications, such as spam filtering [4], emotion recognition [19] and human activity detection [18]. More recently, covariate shift adaptation is jointly optimized with model robustness [29], fairness [28], and conformal prediction [35].

**Unsupervised model performance evaluation**: Model performance evaluation without labels has received relatively limited attention. Domain-specific models' performance can be estimated via certain statistics, such as confidence score [17], rotation prediction [9] and feature statistics of the datasets sampled from a meta-dataset [10] for image recognition. General model evaluation often relies on different assumptions and accessibility [14, 7, 6, 12, 8, 36]. [7] assumes covariate shift and requires users to provide an approximation (slice) of the shifted features, while [6] needs white-box access to the ML models to train an ensemble as a reference. [12] assumes the label distributions are known, while [8] needs a feature independence structure given the labels. When a small number of labels can be obtained, [36] proposes an active model evaluation approach. To our knowledge, this is the first paper that explicitly models the joint shift of both labels and a few features with provable identification guarantees. Moreover, we do not require access to side information such as model design or metadata.

## 2 Preliminaries and Problem Statement

We start by giving the preliminaries and the problem of estimating and explaining performance shift.

**Prediction tasks and ML models.** In this paper we consider the standard classification task: given a $d$-dimensional feature vector $\boldsymbol{x} \in \mathcal{X} \subseteq \mathbb{R}^d$ from the feature space $\mathcal{X}$, the goal is to predict its associated label $y \in \mathcal{Y}$ in the label space $\mathcal{Y}$. Let $f(\cdot) : \mathbb{R}^d \mapsto \mathcal{Y}$ denote an ML model designed for such a task. For simplicity, we assume that $\mathcal{Y} = \{1, 2, \cdots, L\}$. Given the model's prediction $f(\boldsymbol{x})$ and its true label $y$, its performance is quantified by some loss function $\ell(\cdot, \cdot)$. A popular choice is the standard 0-1 loss: $\ell(a, b) = \mathbb{1}_{a=b}$, which we focus on, but other losses are also applicable.

**Joint distribution shift.** The training and inference data for ML often come from two different distributions, referred to as source domain and target domain. Here, we consider the general case when the joint distribution vary across the source and target domains, and call this *joint distribution shift*. Formally, let $\mathbb{P}_s, \mathbb{P}_t : \mathcal{X} \times \mathcal{Y} \mapsto [0, 1]$ denote the source and target domains, and $p_s, p_t$ be their probability density (or mass) functions. Then joint distribution shift means $p_s(\boldsymbol{x}, y) \neq p_t(\boldsymbol{x}, y)$.

**Problem statement.** Suppose we are given a labeled dataset $D_s \triangleq \{(\boldsymbol{x}^{s,i}, y^{s,i})\}_{i=1}^{n_s}$ from the source distribution $\mathbb{P}_s$, an unlabeled dataset $D_t \triangleq \{(\boldsymbol{x}^{t,i})\}_{i=1}^{n_t}$ from the target distribution $\mathbb{P}_t$, and an ML model $f(\cdot)$ predicting the associated label given any feature vector $\boldsymbol{x}$. Our goal is to estimate how much performance changes from the source domain to the target domain. More formally, we aim at estimating the performance shift $\Delta \triangleq \mathbb{E}_{(\boldsymbol{x},y)\sim\mathbb{P}_t}[\ell(f(\boldsymbol{x}), y)] - \mathbb{E}_{(\boldsymbol{x},y)\sim\mathbb{P}_s}[\ell(f(\boldsymbol{x}), y)]$. This is challenging as we do not observe labels on the target domain. In many applications, attributing the performance shift to certain features is also desired. Thus, we are also interested in identifying a set of features to explain the performance shift.

## 3 SJS: A Tractable Unification of Label Shift and Sparse Covariate Shift

At a first glance, estimating the performance shift under joint distribution shift without observing labels from target domain seems hopeless: if the marginal feature distributions are identical for both domains, then observing the features alone should give 0 as the estimated performance shift. However, the label distribution given any feature on the target domain is arbitrary, and thus the estimated shift can be arbitrarily bad. In other words, joint distribution shift is not identifiable with no target labels.

To mitigate nonidentifiability, it's necessary to make additional assumptions. The most popular assumptions in literature are label shift [22] and covariate shift [30]. Label shift assumes that only label distribution may change, but the feature distribution given a label remains, i.e., $p_s(\boldsymbol{x}|y) = p_t(\boldsymbol{x}|y)$. On the other hand, covariate shift assumes that feature distribution can shift, but the label

distribution given the features is fixed, i.e., $p_s(y|\boldsymbol{x}) = p_t(y|\boldsymbol{x})$. However, those assumptions disallow simultaneous changes of both features and labels, which often happen in real-world data [21, 27, 34]. To enable joint feature and label estimation which is tractable, we introduce a subclass of joint distribution shift, *Sparse Joint Shift* (SJS), as follows.

**Definition 1** (Sparse Joint Shift (SJS)). *Suppose for an integer $m \le d$ and an index set $\mathcal{I} \subset [d]$ with size at most $m$ (i.e., $|I| \le m$), $p_s(\boldsymbol{x}_{I^c}|\boldsymbol{x}_I, y) = p_t(\boldsymbol{x}_{I^c}|\boldsymbol{x}_I, y)$. Then we say the source and target pair $(p_s, p_t)$ is under $m$-Sparse Joint Shift, or $m$-SJS. Here, $I^c \triangleq [d] - I$. We call $I$ the shift index set.*

Roughly speaking, SJS allows both labels and a few features to shift, but assumes the remaining features' conditional distribution to stay the same. Section 1 gives one example when SJS occurs, and more examples and discussions can be found in the appendix. Next, we will study when this assumption allows tractable performance shift estimation. All proofs are left to the appendix.

## 3.1 When is sparse joint shift identifiable?

Recall that additional assumptions are needed because the general joint distribution shift is not identifiable. However, when $m = d$, $m$-SJS simply becomes general joint distribution shift. Thus, it is worthy understanding when $m$-SJS resolves the identifiability issue. To do so, let us first formally introduce the notation of identifiability.

**Definition 2** (Identifiable). *Suppose the source-target tuple $(p_s, p_t)$ is under $m$-SJS. $(p_s, p_t)$ is identifiable if and only if for any alternative distribution $p_a(\boldsymbol{x}, y)$, if $p_a(\boldsymbol{x}) = p_t(\boldsymbol{x})$ and $\exists \mathcal{J} \subset [d], |\mathcal{J}| \le m$, such that $p_a(\boldsymbol{x}_{\mathcal{J}^c}|\boldsymbol{x}_{\mathcal{J}}, y) = p_s(\boldsymbol{x}_{\mathcal{J}^c}|\boldsymbol{x}_{\mathcal{J}}, y)$, then $p_a(\boldsymbol{x}, y) = p_t(\boldsymbol{x}, y)$.*

The identifiability can be easily interpreted in words: If a joint feature and label distribution matches the target feature distribution and satisfies the $m$-SJS requirement together with the source distribution, it has to be the target distribution. The following statement shows when $(p_s, p_t)$ is identifiable.

**Theorem 1.** *Suppose $(p_s, p_t)$ is under $m$-SJS. Assume for any set $\mathcal{J} \subset [d], |\mathcal{J}| \le m$ and any fixed $\bar{\boldsymbol{x}} \in \mathcal{X}$, the probability density (or mass) functions $\{p_s(\boldsymbol{x}_{J^c \cap I^c}, \boldsymbol{x}_{\mathcal{J} \cup I} = \bar{\boldsymbol{x}}_{\mathcal{J} \cup I}, y = i)\}_{i=1}^L$ are linearly independent. Then $(p_s, p_t)$ is identifiable.*

This statement sheds light on why uniquely identifying the target distribution without target label is feasible under sparse joint shift. Roughly speaking, $m$-SJS requires that given the shifted features and labels, the remaining features' distribution remains the same on both domains. If those remaining features are different enough (linear independence), they can uniquely determine the distribution of the shifted features and labels. We stress that the linear independence is necessary: if it does not hold, then for any $m$, we can always find some source-target pair $(p_s, p_t)$ which is not identifiable. Linear independence implicitly requires sparsity: if $m > d/2$, then $J^c \cap I^c$ can be empty and the linear independence does not hold. In other words, the sparsity is necessary for the shift to be identifiable.

## 3.2 How does SJS relate to label shift and covariate shift?

A natural question is how does SJS relates to standard label shift and covariate shift. To answer this, let us first introduce label and sparse covariate shift formally.

**Definition 3.** *The source and target $(p_s, p_t)$ is under Label Shift iff $p_s(\boldsymbol{x}|y) = p_t(\boldsymbol{x}|y)$, and under $m$-Sparse Covariate Shift iff $p_s(\boldsymbol{x}_{I^c}, y|\boldsymbol{x}_I) = p_t(\boldsymbol{x}_{I^c}, y|\boldsymbol{x}_I)$ for some index set $I$ with size $m < d$.*

Now we are ready to answer the above question.

**Theorem 2.** *If $(p_s, p_t)$ is under label shift, then it is also under 0-SJS. If $(p_s, p_t)$ is under $m$-sparse covariate shift, then it is also under $m$-SJS. In addition, there exists $(p_s, p_t)$ under $m$-SJS such that it is under neither label shift or covariate shift.*

There are several takeaways. First, label shift implies SJS without additional requirements. In fact, as certain distribution pairs are under SJS but not label shift, SJS is strictly more general than label shift. Second, SJS also includes sparse covariate shift. When $m = d$, SJS completely unifies both label shift and covariate shift, though it is not identifiable. Identifiable SJS, on the other hand, unifies label shift and sparse covariate shift. Finally, SJS also allows shifts not covered by label shift and covaraite shift: the correlation between label and (a set of) features can be shifted.

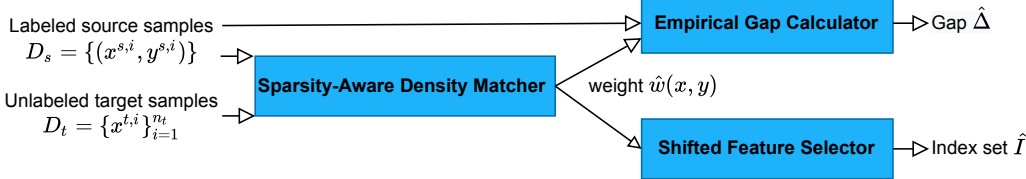

Figure 2: How SEES works. Given labeled source and unlabeled target data, SEES uses a sparsity-aware density matcher to learn a weight function $\hat{w}(\bar{\boldsymbol{x}}, y)$. Next, an empirical gap calculator computes the performance gap $\hat{\Delta}$ by weighing the source samples with the learned $\hat{w}(\bar{\boldsymbol{x}}, y)$. The shifted feature selector extracts the features $\hat{I}$ on which the weight function depends heavily.

## 4 Shift Estimation and Explanation under Sparse Joint Shift

Now we present SEES, an algorithmic framework to estimate and explain the performance shift $\Delta$ when the source and target domain is under $m$-SJS. As shown in Figure 2, it consists of three components: a sparsity-aware density matcher, an empirical gap calculator, and a shifted feature selector. Given the labeled source samples and unlabeled target samples, we first adopt the sparsity-aware density matcher to obtain an estimated ratio of the target and source density functions, denoted by $\hat{w}(\boldsymbol{x}, y)$. Next, the empirical gap calculator is responsible to estimate the performance shift $\Delta$ via appropriately reweighting source samples with ratio $\hat{w}(\boldsymbol{x}, y)$. Finally, the shifted feature selector picks a set of features as the explanation for the shift. We explain each component as follows.

### 4.1 Sparsity-aware density matcher

A key component of SEES is the sparsity-aware density matcher. Here we want to find some weight function $\hat{w}(\boldsymbol{x}, y)$ to induce an estimated target distribution $\hat{p}_t(\boldsymbol{x}, y) \triangleq \hat{w}(\boldsymbol{x}, y) \cdot p_s(\boldsymbol{x}, y)$. Our goals include (i) a small distance between the estimated target distribution and the true target distribution, (ii) $m$-SJS between the source and the estimated target distributions, and (iii) flexible parameterization of the weight function. To achieve those goals, we propose the following optimization framework

$$\min_{w(\boldsymbol{x}, y) \in \mathcal{W}} D(p_t(\boldsymbol{x}), \hat{p}_t(\boldsymbol{x}))$$

$$\text{s.t. } \hat{p}_t(\boldsymbol{x}) = \sum_{y=1}^{L} w(\boldsymbol{x}, y) \cdot p_s(\boldsymbol{x}, y), \text{ and } w(\boldsymbol{x}, y) \text{ depends on at most } m \text{ features of } \boldsymbol{x}. \tag{4.1}$$

Here, $D(\cdot, \cdot)$ is some distance metric that measures the difference between two density functions. We minimize the distance between the induced feature density $\hat{p}_t(\boldsymbol{x})$ and the target feature density $p_t(\boldsymbol{x})$. The minimization is not over joint label and feature distributions since target labels are not available. The induced feature density function can be easily derived from source density function and the weight function, encoded in the first constraint. $m$-SJS is enforced by the second constraint: $m$-SJS means given $m$ features and labels, the distributions of remaining features are fixed across source and the induced domain, which holds if and only if their density ratio $w(\boldsymbol{x}, y)$ only depends on those $m$ features. $\mathcal{W}$ represents the set of all feasible weight functions. Different parameterization can be easily realized by adopting different $\mathcal{W}$. Assume access to density functions $p_s(\boldsymbol{x}, y)$ and $p_t(\boldsymbol{x})$, and a weight function set $\mathcal{W}$ containing the true weight $w^*(\boldsymbol{x}, y) \triangleq \frac{p_t(\boldsymbol{x}, y)}{p_s(\boldsymbol{x}, y)}$. One can easily show the above optimization returns the true weight function $w^*(\boldsymbol{x}, y)$ for identifiable $m$-SJS.

One benefit of the above framework is the flexibility of concrete instantiations. Different design choices, including the distance metric $D(\cdot, \cdot)$ and the weight parameterization space $\mathcal{W}$, can fit different feature types, incorporate domain knowledge, and tradeoff different sample and computational complexity. We give two instantiations of the above optimization: SEES-c for continuous features, and SEES-d for discrete features.

**SEES-c: SEES for continuous features.** For continuous features, we use KL-divergence $D_{KL}(\cdot, \cdot)$ as the distance metric, and initialize the parameterization space $\mathcal{W}$ by linear combinations of $K$ fixed basis functions, $\phi_1(\boldsymbol{x}, y), \phi_2(\boldsymbol{x}, y), \cdots, \phi_K(\boldsymbol{x}, y)$. That is to say, $\mathcal{W} = \{w(\boldsymbol{x}, y) | w(\boldsymbol{x}, y) =$

$\sum_{k=1}^{K} a_{k,y}\phi_k(\boldsymbol{x}, y), a_{k,y} \geq 0, \mathbb{E}_{(\boldsymbol{x},y)\sim\mathbb{P}_s}\left[\sum_{k=1}^{K} a_{k,y}\phi_k(\boldsymbol{x}, y)\right] = 1, \alpha_{k,y} \geq 0\}$. The last two constraints ensure $w(\boldsymbol{x}, y) \cdot p_s(\boldsymbol{x}, y)$ is a valid probability density. Those basis functions encode users' prior knowledge about the shift. A simple choice, for example, is linear functions (when $\boldsymbol{x}_k \geq 0$): setting $K = d$ and $\phi_k(\boldsymbol{x}, y) = \boldsymbol{x}_k$. To model the dependence on different features, let $e_i$ denote all indexes $k$ such that $\phi_k(\cdot)$ depends on feature $\boldsymbol{x}_i$, and introduce a vector $\boldsymbol{\beta} \in \mathbb{R}^d$ such that $\boldsymbol{\beta}_i \triangleq \sqrt{\sum_{k\in e_i}\sum_{y=1}^{L} a_{k,y}^2}$. The feature dependence requirement in Problem 4.1 is equivalent to sparsity constraint $\|\boldsymbol{\beta}\|_0 \leq s$. We can relax the 0-norm by 1-norm and obtain one instantiation as

$$\max_{a_{1,1},a_{1,2},\cdots,a_{K,L}} \mathbb{E}_{\boldsymbol{x}\sim\mathbb{P}_t}\left[\log\sum_{y=1}^{L} p_s(y|\boldsymbol{x})\sum_{k=1}^{K} a_{k,y}\phi_k(\boldsymbol{x}, y)\right] + \eta\sum_{i=1}^{d}\sqrt{\sum_{k\in e_i}\sum_{y=1}^{L} a_{k,y}^2}$$

$$s.t.\ \mathbb{E}_{(\boldsymbol{x},y)\sim\mathbb{P}_s}\left[\sum_{k=1}^{K} a_{k,y}\phi_k(\boldsymbol{x}, y)\right] = 1, \alpha_{k,y} \geq 0$$

where $\eta > 0$ controls the trade-off between sparsity and the KL distance. One benefit of this instantiation is computational efficiency: the constraint is linear in the optimization variables, and the objective is convex. Thus, the problem is convex and can be efficiently solved. The label distribution given feature $p_s(y|\boldsymbol{x})$ is unknown but can be approximated by the ML model $f(\cdot)$ trained on the source domain. Given finite samples, the expectations can be replaced by their empirical estimation.

**SEES-d: SEES for discrete features.** Another interesting instantiation exists for discrete features. With no prior knowledge, we parameterize $\mathcal{W}$ to include all possible $m$-SJS: specifically, $\mathcal{W}$ contains all tuple $(J, w_J(\boldsymbol{x}_J, y))$, where index set $J \subset [d], |J| = m$ represents the shifted features, and weight function $w_J(\boldsymbol{x}_J, y)$ only depends on $\boldsymbol{x}_J$ and $y$.

Features only take finite values, so we can view the density (mass) functions as vectors with finite dimensions. Thus, we adopt the squared $\ell_2$ distance, i.e., $D(\boldsymbol{z}, \boldsymbol{z}') = \sum_{i=1}^{|\boldsymbol{z}|}(\boldsymbol{z}_i - \boldsymbol{z}'_i)^2$ to measure distance. However, naively measuring $\ell_2$ distance between $p_t(\boldsymbol{x})$ and $\hat{p}_t(\boldsymbol{x})$ leads to a computational complexity exponential in $d$. Instead, we measure the distance on a set of marginal densities: given an index set $J$, for every index set with size $2s$ that contains $J$, denoted by $\kappa$, we measure the squared $\ell_2$ distance between $p_t(\boldsymbol{x}_\kappa, f(\boldsymbol{x}))$ and $\hat{p}_t(\boldsymbol{x}_\kappa, f(\boldsymbol{x}))$, and then aggregate over $\kappa$. This design leads to the following instantiation of Problem 4.1

$$\min_{J,w_J(\boldsymbol{x},y)} \sum_{\kappa:J\subseteq\kappa,|\kappa|=2m} \sum_{\bar{f}=1}^{L}\sum_{\bar{\boldsymbol{x}}_\kappa\in\mathcal{X}_\kappa} \|p_t(\bar{\boldsymbol{x}}_\kappa, \bar{f}) - \sum_{\bar{y}=1}^{L} w_J(\bar{\boldsymbol{x}}_J, \bar{y})\cdot p_s(\bar{\boldsymbol{x}}_\kappa, \bar{f}, \bar{y})\|_2^2, \ s.t.|J| = m \quad (4.2)$$

where $p_t(\bar{\boldsymbol{x}}_\kappa, \bar{f})$ and $p_s(\bar{\boldsymbol{x}}_\kappa, \bar{f}, \bar{y})$ are short for $p_t(\boldsymbol{x}_\kappa = \bar{\boldsymbol{x}}_\kappa, f(\boldsymbol{x}) = \bar{f})$ and $p_s(\boldsymbol{x}_\kappa = \bar{\boldsymbol{x}}_\kappa, f(\boldsymbol{x}) = \bar{f}, y = \bar{y})$, respectively.

Compared to the naive approach, the above formulation is much more computationally efficient: the number of parameters in the above objective is only polynomial in the feature dimension $d$. For fixed $J$, the problem is simply a linear regression over the weight $w_J(\boldsymbol{x}_J, y)$ and thus can be efficiently solved. In practice, one can estimate $p_t(\bar{\boldsymbol{x}}_\kappa, \bar{f})$ and $p_s(\bar{\boldsymbol{x}}_\kappa, \bar{f}, \bar{y})$ via labeled source and unlabeled target samples, and then solve the empirical version of the above problem. Compared to using KL-divergence, solving the empirical version produces the correct shifted index set and a weight function close to the true weight $w^*(\boldsymbol{x}, y)$ (under mild conditions). This is formally stated as follows.

**Theorem 3.** *Consider when all features are discrete, i.e., for each $i$, $\boldsymbol{x}_i \in \{1, 2, \cdots, v\}$. Suppose (i) the source and target are under exact $m$-SJS, (ii) for any set $\mathcal{J} \subset [d], |\mathcal{J}| \leq m$ and any $\bar{\boldsymbol{x}} \in \mathcal{X}$, the marginal probability density (or mass) functions $\{p_s(f(\boldsymbol{x}), \boldsymbol{x}_{\mathcal{J}\cup I} = \bar{\boldsymbol{x}}_{\mathcal{J}\cup I}, y = i)\}_{i=1}^{d}$ are linearly independent, and (iii) $w(\boldsymbol{x}, y)$ is bounded by a constant $M$. Then there exists some constant $c$ (independent of $d$, $n_s$ and $n_t$), such that if $\sqrt{\frac{1}{2n_s}} + LM\sqrt{\frac{1}{2n_t}} < c/\sqrt{2m\log d + m\log v + 2\log L + \log 1/\delta}$, then with probability $1 - \delta$, (i) the index set $\hat{J}$ learned by Problem 4.2 matches the true shift index set $I$, and (ii) the produced weights $w_{\hat{j}}(\boldsymbol{x}_{\hat{j}}, y)$ satisfies $\left|w_{\hat{j}}(\boldsymbol{x}_{\hat{j}}, y) - w^*(\boldsymbol{x}, y)\right| \leq O\left(\sqrt{2m\log d + m\log v + 2\log L + \log 1/\delta}\left(\sqrt{\frac{1}{2n_s}} + LM\sqrt{\frac{1}{2n_t}}\right)\right)$.*

Roughly speaking, this statement ensures that, when source and target sample sizes are large enough, with high probability, the true shift index set can be identified with finite samples, and the error rate of the learned weight function is approximately the inverse of sample sizes' square root.

**Comparisons of SEES-c and SEES-d.**  SEES-d enjoys mathematical guarantees, but SEES-c can be computationally more efficient. In practice, we can discretize continuous features to use SEES-d.

### 4.2  Empirical gap calculator and shifted feature selector

Now we explain how the other two components of SEES work. The empirical gap calculator computes the performance shift $\hat{\Delta}$ via three steps. First, it estimates the source performance by $\frac{1}{n_s} \sum_{i=1}^{n_s} \ell(\boldsymbol{x}^{s,i}, y^{s,i})$. Next, it estimates the performance on the induced target distribution. Note that the performance on the induced target domain is $\int \hat{p}_t(\boldsymbol{x}, y)\ell(\boldsymbol{x}, y)d\boldsymbol{x}dy = \int \hat{w}(\boldsymbol{x}, y)p_s(\boldsymbol{x}, y)\ell(\boldsymbol{x}, y)d\boldsymbol{x}dy = \mathbb{E}_{(\boldsymbol{x}, y)\sim \mathbb{P}_s}[\hat{w}(\boldsymbol{x}, y)\ell(\boldsymbol{x}, y)]$. Thus, we use the weighted loss on the source samples $\frac{1}{n_s} \sum_{i=1}^{n_s} \hat{w}(\boldsymbol{x}^{s,i}, y^{s,i})\ell(\boldsymbol{x}^{s,i}, y^{s,i})$ as the estimation. Finally, their difference, i.e., $\hat{\Delta} = \frac{1}{n_s} \sum_{i=1}^{n_s} (\hat{w}(\boldsymbol{x}^{s,i}, y^{s,i}) - 1)\ell(\boldsymbol{x}^{s,i}, y^{s,i})$ is returned as the estimated performance shift.

The shifted feature selector picks a set of features as the shift explanation. For discrete data, the weight function $\hat{w}(\boldsymbol{x}, y)$ learned by the density matcher's instantiation is parameterised as a shifted index $\hat{J}$ and the corresponding weight $\hat{w}_{\hat{j}}(\boldsymbol{x}_{\hat{j}}, y)$. Thus, a natural choice is to return $\hat{I} = \hat{J}$ as the explanation. For continuous data, the weight function is $\hat{w}(\boldsymbol{x}, y) = \sum_{k=1}^{K} \hat{a}_{k,y}\phi_k(\boldsymbol{x}, y)$, where $\hat{a}_{k,y}$ is learned by the corresponding instantiation. Recall that $e_i$ denotes all basis functions that depends on feature $i$. Then $\hat{\boldsymbol{\beta}}_i \triangleq \sqrt{\sum_{k\in e_i} \sum_{y=1}^{L} \hat{a}_{k,y}^2}$ can be viewed as the total contribution of feature $i$. Thus, a simple choice is to pick features with the $m$ largest contributions. Formally, we use $\hat{I} = \{i|\hat{\boldsymbol{\beta}}_i > \hat{\boldsymbol{\beta}}_{(d-m)}\}$, where $\hat{\boldsymbol{\beta}}_{(d-m)}$ is the $d-m$ smallest value in $\hat{\boldsymbol{\beta}}$.

## 5  Experiments

In this section, we study the performance of SEES on several real world datasets with synthetic and natural distribution shifts. Our goal is four-fold: (i) understand when and how SEES estimates the performance shift, (ii) evaluate the trade-offs between the estimation performance reached by SEES and the required dataset sizes, (iii) explore the effects of shift sparsity on SEES' performance, and (iv) validate the effectiveness of SEES on datasets with real world distribution shifts.

**Datasets, ML models and baselines.**  Six datasets are used for evaluation purposes. we first simulate various SJS on BANKCHURN [1], COVID-19 [2], and CREDIT [39] to systematically understand the performance of SEES. Next, we apply SEES on EMPLOY, INCOME, and INSURANCE [11] with real world distribution shifts and perform an in-depth analysis. We use a gradient boosting tree model as the ML model, and results for more models can be found in the Appendix. For comparison, we adopt two state-of-the-art methods for comparison: BBSE [22] for label shift and KLIEP [32] for covariate shift. More details on the experiments can be found in the Appendix.

**Case study.**  We start with a case study on the dataset COVID-19. The task is to predict whether a person tests positive for COVID-19. We simulate a joint shift of the feature aged and label. Specifically, both source and target data contain 5000 young and aged individuals. The positive rate is 40% for both young and aged group from the source. In the target data, we raise the positive rate to 80% for aged group and 50% for young group. This simulates a shift due to a COVID variant more harmful to the elder than its ancestor. We adopt SEES-d as all features are categorical.

Figure 3 summarizes this case study. First note that identifying which feature is shifted is not obvious. As shown in Figure 3(a), the marginal distribution of most features except age and gender has changed from the source to the target. The actual joint shift (Figure 3(b)) is, on the other hand, due to age group and the labels. Identifying shifted age is challenging as the label on the target (last red bar in Figure 3(a)) also shifts but cannot be observed. On the other hand, SEES-d correctly identifies the shifted feature age, and produces a weight function close to the true weight (Figure 3(b) and (c)). This is primarily because SEES-d explicitly exploits the joint shift modeled by SJS. In fact, SEES-d's performance is significantly better than existing methods. As shown in Figure 3(d), the

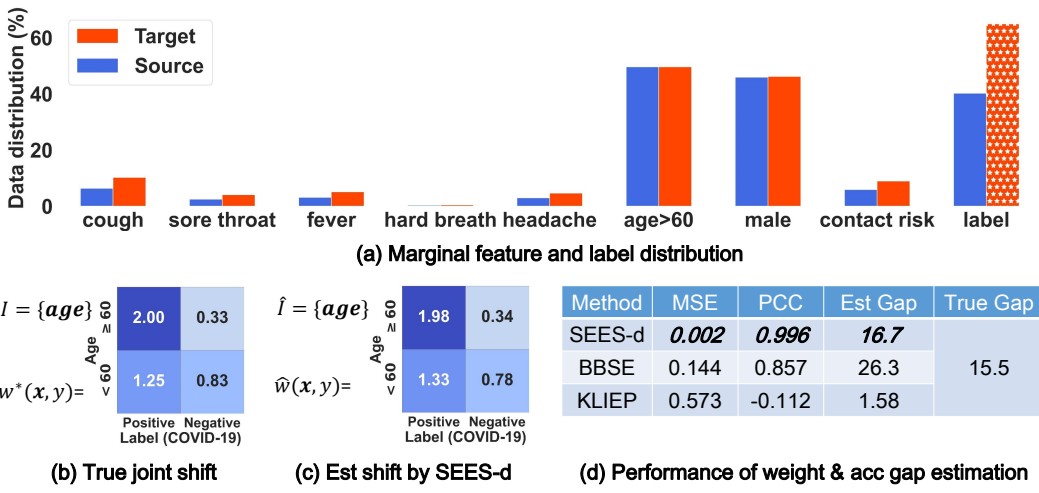

Figure 3: A case study on the COVID-19 dataset. (a) The marginal distribution of labels and all features. The label on the target domain (the last red bar) is not observable. (b) The actual joint shift between source and target data. (c) The mean square error (MSE), pearson correlation coefficient (PCC) between learned and true weights, and the the estimated accuracy gap. Overall, SEES-d significantly improves estimation performance over existing methods.

mean square error (MSE) between the true weights and learned weights is only 0.002 when adopting SEES-d, but 0.144 and 0.573 when using BBSE and KLIEP, respectively. The Pearson correlation coefficient (PCC) between the true weights and weights learned by SEES-d is 0.996, indicating a strong correlation. The weight estimation performance directly affects how precise the estimated accuracy gap is. The estimated gap $\hat{\Delta}$ of SEES-d is 16.7%, which is close to the true gap (15.5%). By contrast, BBSE tends to overestimate the gap (26.3$) while KLIEP underestimates it (1.58%).

**Trade-offs between estimation performance and sample size.** Next, we study the trade-offs between estimation performance and the number of available samples. For simplicity, we simulate various sparse joint shifts with $s = 1$ via (i) specifying marginal distribution of the shifted feature and labels first, and (ii) then drawing random samples from the original dataset conditional on values of specified labels and shifted feature. Same number of samples are allocated to both source and target datasets. Figure 4 shows the simulated data shift (column 1), the squared $\ell_2$ loss of accuracy estimation (column 2), weight estimation (column 3), and the shifted feature discovery rate (column 4) for three datasets. Overall, we observe that the estimation error of SEES-c and SEES-d diminishes as the number of samples increases, while that of BBSE and KLIEP is almost flat.

Table 1: Root mean square error of estimated accuracy gap (%) under real shifts for a gradient boosting model. The numbers are averaged over all source-target pairs in each dataset. Results for other models (e.g., a neural network) can be found in the Appendix. For each dataset and ML model, SEES provides significant estimation error reduction over baselines.

| EMPLOY | | | | INCOME | | | | INSURANCE | | | |
|---|---|---|---|---|---|---|---|---|---|---|---|
| SEES-c | SEES-d | BBSE | KLIEP | SEES-c | SEES-d | BBSE | KLIEP | SEES-c | SEES-d | BBSE | KLIEP |
| **2.90** | 3.00 | 5.20 | 5.20 | **1.90** | 2.40 | 3.00 | 3.40 | **1.70** | 2.20 | 2.10 | 5.00 |

**Effects of shift sparsity.** We have focused on 1-SJS for simplicity, but how the sparsity $s$ affects the estimation performance remains unknown. To answer this, we fix the number of samples to be 10,000, and then measure the performance of all compared methods for joint distribution of label and different number of features. Overall, we observe that the estimation error often grows as more features shift jointly with the labels. This is because more shifted features often implies higher complexity in the distribution shifts and thus more parameters to estimate. It is worthy noting that

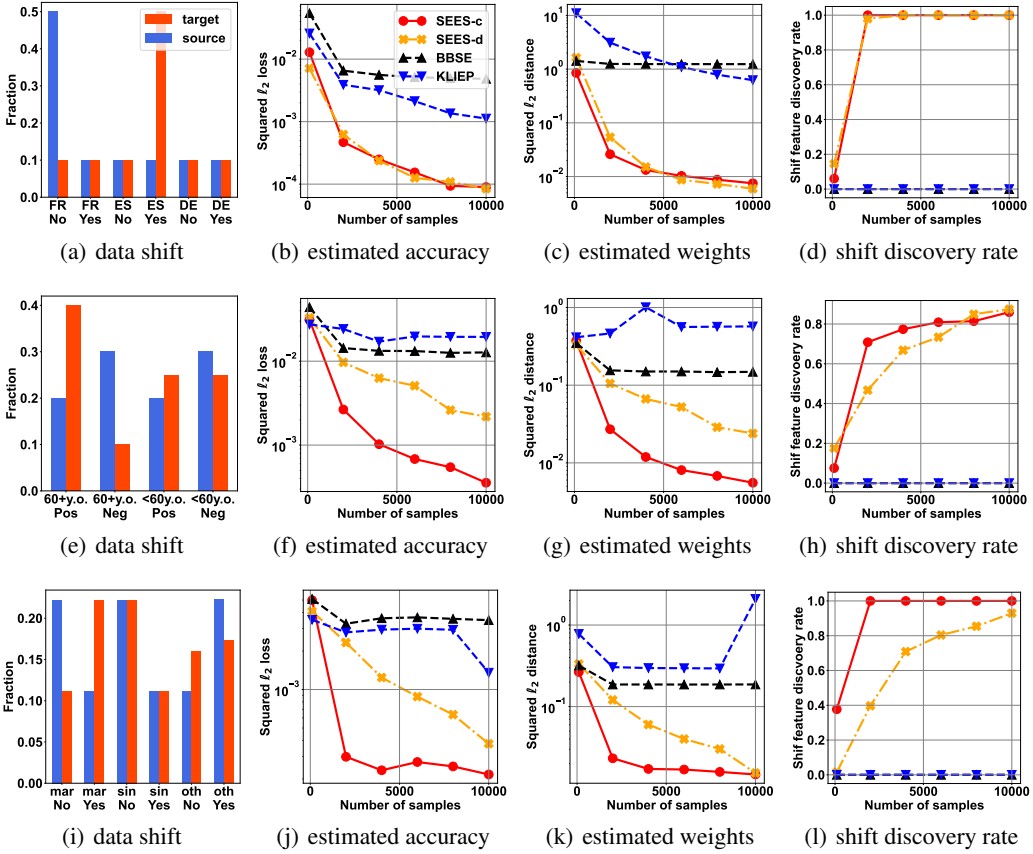

Figure 4: Trade-offs between shift estimation and sample size. We vary the shifted features (first column), and measure performance of the estimated accuracy shift (second column), estimated weights (third column), and how often the true shifted features are discovered (last column). The first, second, and third row corresponds to dataset BANKCHURN, COVID-19, and CREDIT, respectively. Overall, both SEES-c and SEES-d consistently outperforms existing estimation approaches.

$0$-SJS degenerates to label shift, and the performance of SEES-c and SEES-d is slightly worse than BBSE (designed for label shift) under $0$-SJS. More details can be found in the appendix.

**Accuracy gap estimation on real shifts.** Finally we validate the effectiveness of SEES on accuracy estimation with real world shifts. EMPLOY and INCOME are partitioned by geography (states) and INSURANCE is partitioned by time (year). For each partition pair, we train a gradient boosting model on one and estimate its performance on the other. Table 1 shows the estimation error averaged over all partition pairs for each dataset. SEES consistently outperforms BBSE and KLIEP, and reduces the estimation error by up to 66% (1-1.7/5.0). We also evaluate other models (including a neural network) and observe similar results. More results can be found in the Appendix.

## 6 Conclusion

In this paper, we propose Sparse Joint Shift (SJS), a new distribution shift model that accounts for both label and covariate shifts. We show how SJS unifies and generalizes existing distribution shift models and remains identifiable under reasonable assumptions. We develop SEES, an algorithmic framework for unsupervised model performance estimation and explanation under SJS. Both theoretical analysis and empirical study validates the effectiveness of SEES. Our work contributes to making ML more reliable when data can change. A natural next step is how to improve estimation performance under SJS when a small number of target labels can be queried. To stimulate more research on SJS, we also release our code in `https://github.com/stanford-futuredata/SparseJointShift`.

## Acknowledgement

This work was supported in part by a Google PhD Fellowship, a Sloan Fellowship, NSF CCF 1763191, NSF CAREER AWARD 1651570 and 1942926, NIH P30AG059307, NIH U01MH098953, grants from the Chan-Zuckerberg Initiative, Sutherland, and affiliate members and other supporters of the Stanford DAWN project, including Meta, Google, and VMware. We also thank anonymous reviewers for helpful discussion and feedback.

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
