## A  Broader Impact and Limitation Discussion

Monitoring, estimating, and explaining performance of deployed ML models is a growing area with significant economic and social impact. In this paper, we propose SJS, a new data distribution shift model to consider when both labels and features shift after model deployment. We show how SJS generalizes existing data shift models, and further propose SEES, a generic framework that efficiently explains and estimates an ML model's performance under SJS. This may serve as a monitoring tool to help ML practitioners recognize performance changes, discover potential fairness issues and take appropriate business decisions (e.g., switching to other models or debugging the existing ones). One limitation in general is adaption to continuously changing data streams. An online algorithm for performance estimation and explanation under SJS is in need to address this challenge. We will also open source our prototype of SEES serving as a resource for broad community to use.

## B  Missing Proofs

We provide all missing proofs in this section.

### B.1  Proof of Theorem 1

*Proof.* We prove the statement by contradiction. Suppose not. That is to say, there exists a distribution $p_a(\boldsymbol{x}, y)$, such that $p_a(\boldsymbol{x}) = p_t(\boldsymbol{x})$, and there exists some set $\mathcal{J} \subset [d], |\mathcal{J}| \leq m$, such that $p_a(\boldsymbol{x}_{\mathcal{J}^c}|\boldsymbol{x}_{\mathcal{J}}, y) = p_s(\boldsymbol{x}_{\mathcal{J}^c}|\boldsymbol{x}_{\mathcal{J}}, y)$ and $p_a(\boldsymbol{x}, y) \neq p_t(\boldsymbol{x}, y)$. Let $w_a(\boldsymbol{x}, y) \triangleq \frac{p_a(\boldsymbol{x}, y)}{p_s(\boldsymbol{x}, y)}$ denote the ratio between this alternative distribution $p_a$ and the source distribution. Recall that $w^*(\boldsymbol{x}, y) = \frac{p_t(\boldsymbol{x}, y)}{p_s(\boldsymbol{x}, y)}$. To show the contradiction, we simply need to show $p_a(\boldsymbol{x}, y) = p_t(\boldsymbol{x}, y)$, which is equivalent to show $w_a(\boldsymbol{x}, y) = w^*(\boldsymbol{x}, y)$.

By the $m$-SJS assumption, $p_t(\boldsymbol{x}_I^c|\boldsymbol{x}_I, y) = p_s(\boldsymbol{x}_I^c|\boldsymbol{x}_I, y)$, we have

$$w^*(\boldsymbol{x}, y) = \frac{p_t(\boldsymbol{x}, y)}{p_s(\boldsymbol{x}, y)} = \frac{p_t(\boldsymbol{x}_I, y)p_t(\boldsymbol{x}_I^c|\boldsymbol{x}_I, y)}{p_s(\boldsymbol{x}_I, y)p_s(\boldsymbol{x}_I^c|\boldsymbol{x}_I, y)} = \frac{p_t(\boldsymbol{x}_I, y)}{p_s(\boldsymbol{x}_I, y)}$$

Similarly, by the assumption $p_a(\boldsymbol{x}_J^c|\boldsymbol{x}_J, y) = p_s(\boldsymbol{x}_J^c|\boldsymbol{x}_J, y)$, we have

$$w_a(\boldsymbol{x}, y) = \frac{p_a(\boldsymbol{x}, y)}{p_s(\boldsymbol{x}, y)} = \frac{p_a(\boldsymbol{x}_J, y)p_a(\boldsymbol{x}_J^c|\boldsymbol{x}_J, y)}{p_s(\boldsymbol{x}_J, y)p_s(\boldsymbol{x}_J^c|\boldsymbol{x}_I, y)} = \frac{p_a(\boldsymbol{x}_J, y)}{p_s(\boldsymbol{x}_J, y)}$$

Thus, we only need to show

$$\frac{p_t(\boldsymbol{x}_I, y)}{p_s(\boldsymbol{x}_I, y)} = \frac{p_a(\boldsymbol{x}_J, y)}{p_s(\boldsymbol{x}_J, y)}$$

Our approach is to show that the two ratios all satisfy a system of linear equations, which, however, should have only a unique solution. To see this, let us first note that, for any $\bar{\bar{\boldsymbol{x}}}$, we have

$$p_t(\boldsymbol{x}_{I^c \cap \mathcal{J}^c}, \boldsymbol{x}_{I \cup \mathcal{J}} = \bar{\boldsymbol{x}}_{I \cup \mathcal{J}}) = \sum_{y=1}^{d} p_t(\boldsymbol{x}_{I^c \cap \mathcal{J}^c}, \boldsymbol{x}_{I \cup \mathcal{J}} = \bar{\boldsymbol{x}}_{I \cup \mathcal{J}}, y = \bar{y})$$

$$= \sum_{y=1}^{d} p_t(\boldsymbol{x}_{I^c \cap \mathcal{J}^c}, \boldsymbol{x}_{\mathcal{J}} = \bar{\boldsymbol{x}}_{\mathcal{J}}|\boldsymbol{x}_I = \bar{\boldsymbol{x}}_I, y = \bar{y})p_t(\boldsymbol{x}_I = \bar{\boldsymbol{x}}_I, y = \bar{y})$$

$$= \sum_{y=1}^{d} p_s(\boldsymbol{x}_{I^c \cap \mathcal{J}^c}, \boldsymbol{x}_{\mathcal{J}} = \bar{\boldsymbol{x}}_{\mathcal{J}}|\boldsymbol{x}_I = \bar{\boldsymbol{x}}_I, y = \bar{y})p_t(\boldsymbol{x}_I = \bar{\boldsymbol{x}}_I, y = \bar{y})$$

$$= \sum_{y=1}^{d} p_s(\boldsymbol{x}_{I^c \cap \mathcal{J}^c}, \boldsymbol{x}_{I \cup \mathcal{J}} = \bar{\boldsymbol{x}}_{I \cup \mathcal{J}}, y = \bar{y})\frac{p_t(\boldsymbol{x}_I = \bar{\boldsymbol{x}}_I, y = \bar{y})}{p_s(\boldsymbol{x}_I = \bar{\boldsymbol{x}}_I, y = \bar{y})}$$

Here, the first equation is by the total probability rule, the second equation is by the conditional probability rule, the third equation is by the $s$-SJS assumption, and the last equation is by conditional

probability rule again. Similarly, we can obtain

$$p_a(\boldsymbol{x}_{I^c \cap \mathcal{J}^c}, \boldsymbol{x}_{I \cup \mathcal{J}} = \bar{\boldsymbol{x}}_{I \cup \mathcal{J}}) = \sum_{y=1}^{d} p_a(\boldsymbol{x}_{I^c \cap \mathcal{J}^c}, \boldsymbol{x}_{I \cup \mathcal{J}} = \bar{\boldsymbol{x}}_{I \cup \mathcal{J}}, y = \bar{y})$$

$$= \sum_{y=1}^{d} p_a(\boldsymbol{x}_{I^c \cap \mathcal{J}^c}, \boldsymbol{x}_I = \bar{\boldsymbol{x}}_I | \boldsymbol{x}_J = \bar{\boldsymbol{x}}_J, y = \bar{y}) p_t(\boldsymbol{x}_J = \bar{\boldsymbol{x}}_J, y = \bar{y})$$

$$= \sum_{y=1}^{d} p_s(\boldsymbol{x}_{I^c \cap \mathcal{J}^c}, \boldsymbol{x}_I = \bar{\boldsymbol{x}}_I | \boldsymbol{x}_J = \bar{\boldsymbol{x}}_J, y = \bar{y}) p_t(\boldsymbol{x}_J = \bar{\boldsymbol{x}}_J, y = \bar{y})$$

$$= \sum_{y=1}^{d} p_s(\boldsymbol{x}_{I^c \cap \mathcal{J}^c}, \boldsymbol{x}_{I \cup \mathcal{J}} = \bar{\boldsymbol{x}}_{I \cup \mathcal{J}}, y = \bar{y}) \frac{p_t(\boldsymbol{x}_J = \bar{\boldsymbol{x}}_J, y = \bar{y})}{p_s(\boldsymbol{x}_J = \bar{\boldsymbol{x}}_J, y = \bar{y})}$$

where the first equation is by the total probability rule, the second equation is by the conditional probability rule, the third equation is by the assumption that $p_a(\boldsymbol{x}_{\mathcal{J}^c} | \boldsymbol{x}_J, y) = p_s(\boldsymbol{x}_J | \boldsymbol{x}_J, y)$, and the last equation is by conditional probability rule again. Note that we have assumed that $p_a(\boldsymbol{x}) = p_t(\boldsymbol{x})$. Thus, we must have

$$\sum_{y=1}^{d} p_s(\boldsymbol{x}_{I^c \cap \mathcal{J}^c}, \boldsymbol{x}_{I \cup \mathcal{J}} = \bar{\boldsymbol{x}}_{I \cup \mathcal{J}}, y = \bar{y}) \frac{p_t(\boldsymbol{x}_I = \bar{\boldsymbol{x}}_I, y = \bar{y})}{p_s(\boldsymbol{x}_I = \bar{\boldsymbol{x}}_I, y = \bar{y})}$$

$$= \sum_{y=1}^{d} p_s(\boldsymbol{x}_{I^c \cap \mathcal{J}^c}, \boldsymbol{x}_{I \cup \mathcal{J}} = \bar{\boldsymbol{x}}_{I \cup \mathcal{J}}, y = \bar{y}) \frac{p_a(\boldsymbol{x}_J = \bar{\boldsymbol{x}}_J, y = \bar{y})}{p_s(\boldsymbol{x}_J = \bar{\boldsymbol{x}}_J, y = \bar{y})}$$

which is simply

$$\sum_{y=1}^{d} p_s(\boldsymbol{x}_{I^c \cap \mathcal{J}^c}, \boldsymbol{x}_{I \cup \mathcal{J}} = \bar{\boldsymbol{x}}_{I \cup \mathcal{J}}, y = \bar{y}) \left( \frac{p_t(\boldsymbol{x}_I = \bar{\boldsymbol{x}}_I, y = \bar{y})}{p_s(\boldsymbol{x}_I = \bar{\boldsymbol{x}}_I, y = \bar{y})} - \frac{p_a(\boldsymbol{x}_J = \bar{\boldsymbol{x}}_J, y = \bar{y})}{p_s(\boldsymbol{x}_J = \bar{\boldsymbol{x}}_J, y = \bar{y})} \right) = 0$$

By the assumption that $\{p_s(\boldsymbol{x}_{\mathcal{J}^c \cap \mathcal{K}^c}, \boldsymbol{x}_{\mathcal{J} \cup I} = \bar{\boldsymbol{x}}_{\mathcal{J} \cup I}, y = \bar{y})\}_{y=1}^{d}$ are linearly independent, the above system of equations implies $\frac{p_t(\boldsymbol{x}_I = \bar{\boldsymbol{x}}_I, y = \bar{y})}{p_s(\boldsymbol{x}_I = \bar{\boldsymbol{x}}_I, y = \bar{y})} - \frac{p_a(\boldsymbol{x}_J = \bar{\boldsymbol{x}}_J, y = \bar{y})}{p_s(\boldsymbol{x}_J = \bar{\boldsymbol{x}}_J, y = \bar{y})} = 0$. Note that this holds for any $\bar{\boldsymbol{x}}$. Thus, it is simply

$$\frac{p_t(\boldsymbol{x}_I, y)}{p_s(\boldsymbol{x}_I, y)} = \frac{p_a(\boldsymbol{x}_J, y)}{p_s(\boldsymbol{x}_J, y)}$$

That is to say, $w^*(\boldsymbol{x}, y) = w_a(\boldsymbol{x}, y)$ and thus $p_a(\boldsymbol{x}, y) = p_t(\boldsymbol{x}, y)$, which is a contradiction. Thus, the assumption is incorrect and $(p_s, p_t)$ is identifiable, which finishes the proof. $\qquad\square$

## B.2 Proof of Theorem 2

*Proof.* Proving the first half statement is straightforward: suppose $(p_s, p_t)$ is under label shift. Then by definition, $p_t(\boldsymbol{x} | y) = p_s(\boldsymbol{x} | y)$. That is basically $p_t(\boldsymbol{x}_{[d]} | y, \boldsymbol{x}_\varnothing) = p_s(\boldsymbol{x}_{[d]} | y, \boldsymbol{x}_\varnothing)$, which corresponds to 0-SJS with $I = \varnothing$.

Next we show the proof for the second half. Suppose $(p_s, p_t)$ is under sparse covariate shift, i.e., $p_t(\boldsymbol{x}_{I^c} | \boldsymbol{x}_I) = p_s(\boldsymbol{x}_{I^c} | \boldsymbol{x}_I)$ for some $I$ with size $m < d$, and $p_t(y | \boldsymbol{x}) = p_s(y | \boldsymbol{x})$. Adopting the definition of conditional probability, $p_t(y | \boldsymbol{x}) = p_s(y | \boldsymbol{x})$ can be rewritten as

$$\frac{p_t(y, \boldsymbol{x})}{p_t(\boldsymbol{x})} = \frac{p_s(y, \boldsymbol{x})}{p_s(\boldsymbol{x})}$$

By definition of conditional probability, $p_s(\boldsymbol{x}) = p_s(\boldsymbol{x}_I) \cdot p_s(\boldsymbol{x}_{I^c} | \boldsymbol{x}_I)$ and $p_t(\boldsymbol{x}) = p_t(\boldsymbol{x}_I) \cdot p_t(\boldsymbol{x}_{I^c} | \boldsymbol{x}_I)$, and thus we have

$$\frac{p_t(y, \boldsymbol{x})}{p_t(\boldsymbol{x}_I) \cdot p_t(\boldsymbol{x}_{I^c} | \boldsymbol{x}_I)} = \frac{p_s(y, \boldsymbol{x})}{p_s(\boldsymbol{x}_I) \cdot p_s(\boldsymbol{x}_{I^c} | \boldsymbol{x}_I)}$$

By the assumption $p_t(\boldsymbol{x}_{I^c}|\boldsymbol{x}_I) = p_s(\boldsymbol{x}_{I^c}|\boldsymbol{x}_I)$, we can simplify this as

$$\frac{p_t(y, \boldsymbol{x})}{p_t(\boldsymbol{x}_I)} = \frac{p_s(y, \boldsymbol{x})}{p_s(\boldsymbol{x}_I)}$$

By definition of conditional probability, this is basically

$$p_t(y, \boldsymbol{x}_{I^c}|\boldsymbol{x}_I) = p_s(y, \boldsymbol{x}_{I^c}|\boldsymbol{x}_I)$$

which means $(p_s, p_t)$ is under $m$-SJS.

Now we show the last piece of the statement by construction. Consider the case of $d = 2$. $\boldsymbol{x}_1, \boldsymbol{x}_2, y$ are all binary variables. $\mathbb{P}_s$ is generated as follows: $y$ is first generated from Bernoulli distribution $Bern(0.5)$. If $y = 0$, $\boldsymbol{x}_1, \boldsymbol{x}_2$ are independently generated from $Bern(0.7)$ and $Bern(0.6)$. If $y = 1$, $\boldsymbol{x}_1, \boldsymbol{x}_2$ are independently generated from $Bern(0.1)$ and $Bern(0.2)$. $\mathbb{P}_t$ is generated as follows: $y$ is first generated from $Bern(0.6)$. If $y = 0$, $\boldsymbol{x}_1, \boldsymbol{x}_2$ are independently generated from $Bern(0.5)$ and $Bern(0.6)$. If $y = 1$, $\boldsymbol{x}_1, \boldsymbol{x}_2$ are independently generated from $Bern(0.5)$ and $Bern(0.2)$.

It is easy to see that $(p_s, p_t)$ is under 1-SJS with associated shift index set $I = \{1\}$, since $\boldsymbol{x}_2$ is independent of $\boldsymbol{x}_1$ and only depends on $y$. However,

$$p_t(\boldsymbol{x}_1 = 1|y = 1) = 0.5 \neq 0.1 = p_s(\boldsymbol{x}_1 = 1|y = 1)$$

and thus $(p_s, p_t)$ is not under label shift. In addition,

$$p_t(y = 1|\boldsymbol{x}_1 = 1, \boldsymbol{x}_2 = 1) = \frac{0.6 \times 0.5 \times 0.2}{0.6 \times 0.5 \times 0.2 + 0.4 \times 0.5 \times 0.6} = \frac{1}{3}$$

$$p_s(y = 1|\boldsymbol{x}_1 = 1, \boldsymbol{x}_2 = 1) = \frac{0.5 \times 0.1 \times 0.2}{0.5 \times 0.1 \times 0.2 + 0.5 \times 0.7 \times 0.6} = \frac{1}{22} \neq p_t(y = 1|\boldsymbol{x}_1 = 1, \boldsymbol{x}_2 = 1)$$

and thus $(p_s, p_t)$ is not under covaraite shift, which completes the proof. $\square$

### B.3 Proof of Theorem 3

*Proof.* We prove the statement via three main steps. First, we show that given full access to the distribution, the optimization over the marginal mass functions are sufficient to obtain the correct shifted features and weights. Next, we demonstrate that a large enough number of samples ensures the identified index set stays the same as when full access to the distribution is given with high probability. Finally, we can prove that with high probability, the learned weight function with large number of samples is close to the learned weight function when full distribution is known.

To proceed, let us introduce a few more notations for convenience.

- The distance used by SEES-d: $dd(J, w_J) \triangleq \sum_{\kappa : J \subseteq \kappa, |\kappa| = 2m} \sum_{\bar{f}=1}^{L} \sum_{\bar{\boldsymbol{x}}_\kappa \in \mathcal{X}_\kappa} \|p_t(\bar{\boldsymbol{x}}_\kappa, \bar{f}) - \sum_{\bar{y}=1}^{L} w_J(\bar{\boldsymbol{x}}_J, \bar{y}) \cdot p_s(\bar{\boldsymbol{x}}_\kappa, \bar{f}, \bar{y})\|_2^2$.

- The empirical distance used by SEES-d with finite samples: $\hat{dd}(J, w_J) \triangleq \sum_{\kappa : J \subseteq \kappa, |\kappa| = 2m} \sum_{\bar{f}=1}^{L} \sum_{\bar{\boldsymbol{x}}_\kappa \in \mathcal{X}_\kappa} \|\hat{p}_t(\bar{\boldsymbol{x}}_\kappa, \bar{f}) - \sum_{\bar{y}=1}^{L} w_J(\bar{\boldsymbol{x}}_J, \bar{y}) \cdot \hat{p}_s(\bar{\boldsymbol{x}}_\kappa, \bar{f}, \bar{y})\|_2^2$

- The optimal weight function when the shifted feature set is fixed to $J$: $w_J^* \triangleq \arg\min_{w_J(\boldsymbol{x}, y)} \sum_{\kappa : J \subseteq \kappa, |\kappa| = 2m} \sum_{\bar{f}=1}^{L} \sum_{\bar{\boldsymbol{x}}_\kappa \in \mathcal{X}_\kappa} \|p_t(\bar{\boldsymbol{x}}_\kappa, \bar{f}) - \sum_{\bar{y}=1}^{L} w_J(\bar{\boldsymbol{x}}_J, \bar{y}) \cdot p_s(\bar{\boldsymbol{x}}_\kappa, \bar{f}, \bar{y})\|_2^2$

- The optimal weight function when the shifted feature set is fixed to $J$ and full distribution is available: $w_J^* \triangleq \arg\min_{w_J(\boldsymbol{x}, y)} \sum_{\kappa : J \subseteq \kappa, |\kappa| = 2m} \sum_{\bar{f}=1}^{L} \sum_{\bar{\boldsymbol{x}}_\kappa \in \mathcal{X}_\kappa} \|p_t(\bar{\boldsymbol{x}}_\kappa, \bar{f}) - \sum_{\bar{y}=1}^{L} w_J(\bar{\boldsymbol{x}}_J, \bar{y}) \cdot p_s(\bar{\boldsymbol{x}}_\kappa, \bar{f}, \bar{y})\|_2^2$

- The optimal weight function when the shifted feature set is fixed to $J$ and only finite samples are available: $\hat{w}_J^* \triangleq \arg\min_{w_J(\boldsymbol{x}, y)} \sum_{\kappa : J \subseteq \kappa, |\kappa| = 2m} \sum_{\bar{f}=1}^{L} \sum_{\bar{\boldsymbol{x}}_\kappa \in \mathcal{X}_\kappa} \|\hat{p}_t(\bar{\boldsymbol{x}}_\kappa, \bar{f}) - \sum_{\bar{y}=1}^{L} w_J(\bar{\boldsymbol{x}}_J, \bar{y}) \cdot \hat{p}_s(\bar{\boldsymbol{x}}_\kappa, \bar{f}, \bar{y})\|_2^2$

Next we offer a few useful lemmas before giving the full proof.

**Lemma 4.** *Let $\Omega$ be a compact set, and $\boldsymbol{z}_1^*, \boldsymbol{z}_2^*$ be the optimal solution to the problems*

$$\min_{\boldsymbol{z} \in \Omega} f_1(\boldsymbol{z})$$

*and*

$$\min_{\boldsymbol{z} \in \Omega} f_2(\boldsymbol{z})$$

*where $f_1(\cdot), f_2(\cdot)$ are two functions defined on $\Omega$ such that $|f_1(\boldsymbol{z}) - f_2(\boldsymbol{z})| \leq \Delta, \forall \boldsymbol{z} \in \Omega$. Then we have $|f_1(\boldsymbol{z}_1^*) - f_2(\boldsymbol{z}_2^*)| \leq \Delta$. If $f_1$ is strongly convex with parameter $\lambda$ and $\Omega$ is the full real vector space, then $\|\boldsymbol{z}_1^* - \boldsymbol{z}_2^*\|_2^2 \leq \frac{4\Delta}{\lambda}$.*

*Proof.* We start with the first half statement. By $|f_1(\boldsymbol{z}) - f_2(\boldsymbol{z})| \leq \Delta, \forall \boldsymbol{z} \in \Omega$, we have $f_1(\boldsymbol{z}_1^*) \leq \Delta + f_2(\boldsymbol{z}_1^*)$. Subtracting $f_2(\boldsymbol{z}_2^*)$ from both sides gives

$$f_1(\boldsymbol{z}_1^*) - f_2(\boldsymbol{z}_2^*) \leq \Delta + f_2(\boldsymbol{z}_1^*) - f_2(\boldsymbol{z}_2^*)$$

Observe that $\boldsymbol{z}_2^*$ is the optimal solution to minimizing $f_2(\cdot)$ on the set $\Omega$. Thus, $f_2(\boldsymbol{z}_1^*) \leq f_2(\boldsymbol{z}_2^*)$ must hold. Thus, the above inequality becomes

$$f_1(\boldsymbol{z}_1^*) - f_2(\boldsymbol{z}_2^*) \leq \Delta$$

By symmetry of the two functions, we can obtain

$$f_2(\boldsymbol{z}_2^*) - f_1(\boldsymbol{z}_1^*) \leq \Delta$$

Combining the two inequalities gives

$$|f_2(\boldsymbol{z}_2^*) - f_1(\boldsymbol{z}_1^*)| \leq \Delta$$

Next let us turn to the second half. We first notice that $|f_1(\boldsymbol{z}_1^*) - f_2(\boldsymbol{z}_2^*)| \leq 2\Delta$. To see this, we can decompose the difference $f_1(\boldsymbol{z}_2^*) - f_2(\boldsymbol{z}_1^*)$ as

$$f_1(\boldsymbol{z}_2^*) - f_1(\boldsymbol{z}_1^*) = f_1(\boldsymbol{z}_2^*) - f_2(\boldsymbol{z}_2^*) + f_2(\boldsymbol{z}_2^*) - f_2(\boldsymbol{z}_1^*) + f_2(\boldsymbol{z}_1^*) - f_1(\boldsymbol{z}_1^*)$$

Here, $f_1(\boldsymbol{z}_2^*) - f_2(\boldsymbol{z}_2^*) \leq \Delta$ and $f_2(\boldsymbol{z}_1^*) - f_1(\boldsymbol{z}_1^*) \leq \Delta$ by the assumption $|f_1(\boldsymbol{z}) - f_2(\boldsymbol{z})| \leq \Delta, \forall \boldsymbol{z} \in \Omega$. $\boldsymbol{z}_2^*$ is the optimal solution to minimizing $f_2(\cdot)$, and thus $f_2(\boldsymbol{z}_2^*) - f_2(\boldsymbol{z}_1^*) \leq 0$. Therefore, combining all those leads to

$$f_1(\boldsymbol{z}_2^*) - f_1(\boldsymbol{z}_1^*) = f_1(\boldsymbol{z}_2^*) - f_2(\boldsymbol{z}_2^*) + f_2(\boldsymbol{z}_2^*) - f_2(\boldsymbol{z}_1^*) + f_2(\boldsymbol{z}_1^*) - f_1(\boldsymbol{z}_1^*) \leq 2\Delta$$

Meanwhile, $f_1(\boldsymbol{z}_2^*) - f_1(\boldsymbol{z}_1^*) \geq 0$ as $\boldsymbol{z}_1^*$ is the optimal solution to minimizing $f_1(\cdot)$. That is to say,

$$|f_1(\boldsymbol{z}_2^*) - f_1(\boldsymbol{z}_1^*)| \leq 2\Delta$$

Note that $f_1(\cdot)$ is a strongly convex function with parameter $\lambda$. Thus, we have

$$f_1(\boldsymbol{z}') \geq f_1(\boldsymbol{z}) + < \frac{\partial f_1(\boldsymbol{z})}{\partial \boldsymbol{z}}, \boldsymbol{z}' - \boldsymbol{z} > + \frac{\lambda}{2}\|\boldsymbol{z}' - \boldsymbol{z}\|_2^2$$

for any $\boldsymbol{z}', \boldsymbol{z}$. Now let us set $\boldsymbol{z} = \boldsymbol{z}_1^*, \boldsymbol{z}' = \boldsymbol{z}_2^*$. Remember that $\boldsymbol{z}_1^*$ is the optimal solution to minimizing $f_1(\cdot)$ and $\Omega$ is the full space. Thus, $\frac{\partial f_1(\boldsymbol{z}_1^*)}{\partial \boldsymbol{z}} = 0$. As a result, the above inequality becomes

$$f_1(\boldsymbol{z}_2^*) - f_1(\boldsymbol{z}_1^*) \geq \frac{\lambda}{2}\|\boldsymbol{z}_2^* - \boldsymbol{z}_1^*\|_2^2$$

By $|f_1(\boldsymbol{z}_2^*) - f_1(\boldsymbol{z}_1^*)| \leq 2\Delta$, we obtain

$$2\Delta \geq \frac{\lambda}{2}\|\boldsymbol{z}_2^* - \boldsymbol{z}_1^*\|_2^2$$

Rearranging the terms gives

$$\|\boldsymbol{z}_2^* - \boldsymbol{z}_1^*\|_2^2 \leq \frac{4\Delta}{\lambda}$$

which completes the proof. $\qquad\square$

**Lemma 5.** *Suppose the source and target are under exact $s$-SJS, and for any set $\mathcal{J} \subset [d], |\mathcal{J}| \leq s$ and any $\bar{\boldsymbol{x}} \in \mathcal{X}$, the marginal probability mass functions $\{p_s(f(\boldsymbol{x}), \boldsymbol{x}_{\mathcal{J} \cup I} = \bar{\boldsymbol{x}}_{\mathcal{J} \cup I}, y = \bar{y})\}_{y=1}^d$ are linearly independent. Then $dd(J, w_J) = 0$ if and only if $J = I$ and $w_J(\boldsymbol{x}_J, y) = w^*(\boldsymbol{x}, y)$.*

*Proof.* Let us first relate the marginal density functions on the target domain to those on the source domain. Recall that

$$w^*(\boldsymbol{x}, y) = \frac{p_t(\boldsymbol{x}, y)}{p_s(\boldsymbol{x}, y)}$$

denote the true weights between the target and source density functions. Since the shift is only due to $\boldsymbol{x}_I$, $w^*(\boldsymbol{x}, y)$ only depends on $\boldsymbol{x}_I$. Abusing the notation a little bit, we use $w^*(\boldsymbol{x}_I, y)$ to denote $w^*(\boldsymbol{x}, y)$. Now we can write $p_t(\boldsymbol{x}, y) = w^*(\boldsymbol{x}_I, y) \cdot p_s(\boldsymbol{x}, y)$. Thus, the marginal distribution of $(\boldsymbol{x}_\kappa, f(\boldsymbol{x}))$ on the target domain can be written as

$$p_t(\bar{\boldsymbol{x}}_\kappa, \bar{f}) = \sum_{\bar{y}=1}^{L} \sum_{\boldsymbol{z}:\boldsymbol{z}_\kappa=\bar{\boldsymbol{x}}_\kappa, f(\boldsymbol{z})=\bar{f}} p_t(\boldsymbol{x}=\boldsymbol{z}, y=\bar{y})$$

$$= \sum_{\bar{y}=1}^{L} \sum_{\boldsymbol{z}:\boldsymbol{z}_\kappa=\bar{\boldsymbol{x}}_\kappa, f(\boldsymbol{z})=\bar{f}} w^*(\boldsymbol{z}_I, \bar{y}) \cdot p_s(\boldsymbol{x}=\boldsymbol{z}, y=\bar{y})$$

Now let us consider the two directions of the statement separately.

- $J = I$ and $w_J(\boldsymbol{x}_J, y) = w^*(\boldsymbol{x}, y) \implies dd(J, w_J) = 0$: In this case, $I = J \subseteq \kappa$ and thus $\boldsymbol{z}_I$ is forced to be $\bar{\boldsymbol{x}}_\kappa$. Hence, the marginal distribution becomes

$$p_t(\bar{\boldsymbol{x}}_\kappa, \bar{f}) = \sum_{\bar{y}=1}^{L} \sum_{\boldsymbol{z}:\boldsymbol{z}_\kappa=\bar{\boldsymbol{x}}_\kappa, f(\boldsymbol{z})=\bar{f}} w^*(\boldsymbol{z}_I, \bar{y}) \cdot p_s(\boldsymbol{x}=\boldsymbol{z}, y=\bar{y})$$

$$= \sum_{\bar{y}=1}^{L} w^*(\bar{\boldsymbol{x}}_I, \bar{y}) \sum_{\boldsymbol{z}:\boldsymbol{z}_\kappa=\bar{\boldsymbol{x}}_\kappa, f(\boldsymbol{z})=\bar{f}} p_s(\boldsymbol{x}=\boldsymbol{z}, y=\bar{y})$$

$$= \sum_{\bar{y}=1}^{L} w^*(\bar{\boldsymbol{x}}_I, \bar{y}) p_s(\boldsymbol{x}_\kappa=\bar{\boldsymbol{x}}_\kappa, f(\boldsymbol{x})=\bar{f}, y=\bar{y})$$

$$= \sum_{\bar{y}=1}^{L} w^*(\bar{\boldsymbol{x}}_J, \bar{y}) p_s(\bar{\boldsymbol{x}}_\kappa, \bar{f}, \bar{y})$$

  where the second equation is because $\boldsymbol{z}_I$ is fixed and does not depend on the inner summation, the third is by applying definition of conditional probability, and the last is simply change of notations. The above equation is simply

$$p_t(\bar{\boldsymbol{x}}_\kappa, \bar{f}) - \sum_{\bar{y}=1}^{L} w^*(\bar{\boldsymbol{x}}_J, \bar{y}) p_s(\bar{\boldsymbol{x}}_\kappa, \bar{f}, \bar{y}) = 0$$

  which holds for every $\kappa$. Since $w_J(\boldsymbol{x}_J, y) = w^*(\boldsymbol{x}_J, y)$, it is equivalent to

$$p_t(\bar{\boldsymbol{x}}_\kappa, \bar{f}) - \sum_{\bar{y}=1}^{L} w_J(\bar{\boldsymbol{x}}_J, \bar{y}) p_s(\bar{\boldsymbol{x}}_\kappa, \bar{f}, \bar{y}) = 0$$

  Thus, summing over the square of it also leads to 0, i.e.,

$$dd(J, w_J) \triangleq \sum_{\kappa:J\subseteq\kappa, |\kappa|=2s} \sum_{\bar{f}=1}^{L} \sum_{\bar{\boldsymbol{x}}_\kappa \in \mathcal{X}_\kappa} \left\| p_t(\bar{\boldsymbol{x}}_\kappa, \bar{f}) - \sum_{\bar{y}=1}^{L} w_J(\bar{\boldsymbol{x}}_J, \bar{y}) \cdot p_s(\bar{\boldsymbol{x}}_\kappa, \bar{f}, \bar{y}) \right\|_2^2 = 0.$$

- $dd(J, w_J) = 0 \implies I = J, w_J(\boldsymbol{x}_J, y) = w^*(\boldsymbol{x}, y)$: $dd(J, w_J) = 0$ implies that

$$p_t(\bar{\boldsymbol{x}}_\kappa, \bar{f}) - \sum_{\bar{y}=1}^{L} w_J(\bar{\boldsymbol{x}}_J, \bar{y}) p_s(\bar{\boldsymbol{x}}_\kappa, \bar{f}, \bar{y}) = 0$$

holds for each $\kappa$. In particular, consider $\kappa$ that contains both $I$ and $J$. Then $\boldsymbol{z}_\kappa = \bar{\boldsymbol{x}}_\kappa$ implies $\boldsymbol{z}_I = \bar{\boldsymbol{x}}_I$. Hence, the marginal distribution can be written as

$$p_t(\bar{\boldsymbol{x}}_\kappa, \bar{f}) = \sum_{\bar{y}=1}^{L} \sum_{\boldsymbol{z}:\boldsymbol{z}_\kappa=\bar{\boldsymbol{x}}_\kappa, f(\boldsymbol{z})=\bar{f}} w^*(\boldsymbol{z}_I, \bar{y}) \cdot p_s(\boldsymbol{x} = \boldsymbol{z}, y = \bar{y})$$

$$= \sum_{\bar{y}=1}^{L} w^*(\bar{\boldsymbol{x}}_I, \bar{y}) \sum_{\boldsymbol{z}:\boldsymbol{z}_\kappa=\bar{\boldsymbol{x}}_\kappa, f(\boldsymbol{z})=\bar{f}} p_s(\boldsymbol{x} = \boldsymbol{z}, y = \bar{y})$$

$$= \sum_{\bar{y}=1}^{L} w^*(\bar{\boldsymbol{x}}_I, \bar{y}) p_s(\boldsymbol{x}_\kappa = \bar{\boldsymbol{x}}_\kappa, f(\boldsymbol{x}) = \bar{f}, y = \bar{y})$$

$$= \sum_{\bar{y}=1}^{L} w^*(\bar{\boldsymbol{x}}_I, \bar{y}) p_s(\bar{\boldsymbol{x}}_\kappa, \bar{f}, \bar{y})$$

where the second equation is because $\boldsymbol{z}_I$ is fixed and does not depend on the inner summation, the third is by applying definition of conditional probability, and the last is simply change of notations. Comparing this with the above equation, we end up with

$$\sum_{\bar{y}=1}^{L} w^*(\bar{\boldsymbol{x}}_I, \bar{y}) p_s(\bar{\boldsymbol{x}}_\kappa, \bar{f}, \bar{y}) - \sum_{\bar{y}=1}^{L} w_J(\bar{\boldsymbol{x}}_J, \bar{y}) p_s(\bar{\boldsymbol{x}}_\kappa, \bar{f}, \bar{y}) = 0$$

Or alternatively,

$$\sum_{\bar{y}=1}^{L} [w^*(\bar{\boldsymbol{x}}_I, \bar{y}) - w_J(\bar{\boldsymbol{x}}_J, \bar{y})] p_s(\bar{\boldsymbol{x}}_\kappa, \bar{f}, \bar{y}) = 0$$

which holds for any $\bar{f}, \bar{\boldsymbol{x}}_\kappa$. By the linear independence assumption, this holds if and only if all the coefficients are 0, i.e., $w^*(\bar{\boldsymbol{x}}_I, \bar{y}) - w_J(\bar{\boldsymbol{x}}_J, \bar{y}) = 0$ for all $\bar{\boldsymbol{x}}$ and $\bar{y}$. That is to say, the two functions are identical: $w^*(\boldsymbol{x}_I, y) = w_J(\boldsymbol{x}_J, y)$. As $w_J(\boldsymbol{x}_J, y)$ and $W_*(\boldsymbol{x}_J, y)$ are identical, they can only depend on variables in the set $I \cap J$. That is to say, there exists another importance weights $w_{I \cap J}(\boldsymbol{x}_{I \cap J}, y)$ which results in the same target distribution produced by $W_I(\bar{\boldsymbol{x}}_I, y)$. By the assumption, the shift among the two distribution is exactly $s$-SJS. Hence, we have $|I \cap J| = s$. $|I| = s$ and $I \cap J \subseteq I$ implies $I \cap J = I$ and thus $J \supseteq I$. $|J| = s$ further implies $J = I$.

Therefore, $dd(J, w_J) = 0 \Leftrightarrow I = J, w_J(\boldsymbol{x}_J, y) = w^*(\boldsymbol{x}, y)$, which completes the proof. $\qquad\square$

**Lemma 6.** *With probability at least $1 - \delta$, for any possible $J \subseteq [d], |J| = s$, we have*

$$|dd(J, w_J^*) - \hat{dd}(J, \hat{w}_J^*)|$$

$$\leq 3 \cdot (2s\bar{v})^s ML^2 \sqrt{2s \log d + s \log \bar{v} + 2 \log L + \log 1/\delta} \left( \sqrt{\frac{1}{2n_s}} + LM \sqrt{\frac{1}{2n_t}} \right).$$

*and*

$$\|w_J^* - \hat{w}_J^*\|_2^2 \leq O \left( ML^2 \left( \sqrt{\frac{\log 1/\delta}{2n_s}} + LM \sqrt{\frac{\log 1/\delta}{2n_t}} \right) \right)$$

*Proof.* Let us start by considering a fixed $J$. By definition of $w_J^*$ and $\hat{w}_J^*$, we have

$$dd(J, w_J^*) = \min_{w_J(\boldsymbol{x}, y)} \sum_{\kappa: J \subseteq \kappa, |\kappa|=2s} \sum_{\bar{f}=1}^{L} \sum_{\bar{\boldsymbol{x}}_\kappa \in \mathcal{X}_\kappa} \|p_t(\bar{\boldsymbol{x}}_\kappa, \bar{f}) - \sum_{\bar{y}=1}^{L} w_J(\bar{\boldsymbol{x}}_J, \bar{y}) \cdot p_s(\bar{\boldsymbol{x}}_\kappa, \bar{f}, \bar{y})\|_2^2$$

and

$$\hat{dd}(J, \hat{w}_J^*) = \min_{w_J(\boldsymbol{x}, y)} \sum_{\kappa: J \subseteq \kappa, |\kappa|=2s} \sum_{\bar{f}=1}^{L} \sum_{\bar{\boldsymbol{x}}_\kappa \in \mathcal{X}_\kappa} \|\hat{p}_t(\bar{\boldsymbol{x}}_\kappa, \bar{f}) - \sum_{\bar{y}=1}^{L} w_J(\bar{\boldsymbol{x}}_J, \bar{y}) \cdot \hat{p}_s(\bar{\boldsymbol{x}}_\kappa, \bar{f}, \bar{y})\|_2^2$$

Let us first show that the above two objective functions are close for any fixed $w_J$. To see this, we first apply difference of two squares to obtain

$$\left(\hat{p}_t(\bar{\boldsymbol{x}}_\kappa, \bar{f}) - \sum_{\bar{y}=1}^{L} w_J(\bar{\boldsymbol{x}}_J, \bar{y})\hat{p}_s(\bar{\boldsymbol{x}}_\kappa, \bar{f}, \bar{y})\right)^2 - \left(p_t(\bar{\boldsymbol{x}}_\kappa, \bar{f}) - \sum_{\bar{y}=1}^{L} w_J(\bar{\boldsymbol{x}}_J, \bar{y})p_s(\bar{\boldsymbol{x}}_\kappa, \bar{f}, \bar{y})\right)^2$$

$$= \left(\hat{p}_t(\bar{\boldsymbol{x}}_\kappa, \bar{f}) + p_t(\bar{\boldsymbol{x}}_\kappa, \bar{f}) - \sum_{\bar{y}=1}^{L} w_J(\bar{\boldsymbol{x}}_J, y)\left(\hat{p}_s(\bar{\boldsymbol{x}}_\kappa, \bar{f}, \bar{y}) + p_s(\bar{\boldsymbol{x}}_\kappa, \bar{f}, \bar{y})\right)\right)$$

$$\cdot \left(\hat{p}_t(\bar{\boldsymbol{x}}_\kappa, \bar{f}) - p_t(\bar{\boldsymbol{x}}_\kappa, \bar{f}) - \sum_{\bar{y}=1}^{L} w_J(\bar{\boldsymbol{x}}_J, y)\left(\hat{p}_s(\bar{\boldsymbol{x}}_\kappa, \bar{f}, \bar{y}) - p_s(\bar{\boldsymbol{x}}_\kappa, \bar{f}, \bar{y})\right)\right)$$

(B.1)

Note that all estimated probability mass must be bounded by 1, and by assumption, $|w_J| \leq M$. Thus,

$$\left|\hat{p}_t(\bar{\boldsymbol{x}}_\kappa, \bar{f}) + p_t(\bar{\boldsymbol{x}}_\kappa, \bar{f}) - \sum_{\bar{y}=1}^{L} w_J(\bar{\boldsymbol{x}}_J, y)\left(\hat{p}_s(\bar{\boldsymbol{x}}_\kappa, \bar{f}, \bar{y}) + p_s(\bar{\boldsymbol{x}}_\kappa, \bar{f}, \bar{y})\right)\right| \leq 2 + 2LM \leq 3LM$$

(B.2)

and

$$\left|\hat{p}_t(\bar{\boldsymbol{x}}_\kappa, \bar{f}) - p_t(\bar{\boldsymbol{x}}_\kappa, \bar{f}) - \sum_{\bar{y}=1}^{L} w_J(\bar{\boldsymbol{x}}_J, \bar{y})\left(\hat{p}_s(\bar{\boldsymbol{x}}_\kappa, \bar{f}, \bar{y}) - p_s(\bar{\boldsymbol{x}}_\kappa, \bar{f}, \bar{y})\right)\right|$$

$$\leq \left|\hat{p}_t(\bar{\boldsymbol{x}}_\kappa, \bar{f}) - p_t(\bar{\boldsymbol{x}}_\kappa, \bar{f})\right| + \left|\sum_{\bar{y}=1}^{L} w_J(\bar{\boldsymbol{x}}_J, \bar{y})\left(\hat{p}_s(\bar{\boldsymbol{x}}_\kappa, \bar{f}, \bar{y}) - p_s(\bar{\boldsymbol{x}}_\kappa, \bar{f}, \bar{y})\right)\right|$$

$$\leq \left|\hat{p}_t(\bar{\boldsymbol{x}}_\kappa, \bar{f}) - p_t(\bar{\boldsymbol{x}}_\kappa, \bar{f})\right| + \sum_{\bar{y}=1}^{L} w_J(\bar{\boldsymbol{x}}_J, \bar{y})\left|\left(\hat{p}_s(\bar{\boldsymbol{x}}_\kappa, \bar{f}, \bar{y}) - p_s(\bar{\boldsymbol{x}}_\kappa, \bar{f}, \bar{y})\right)\right|$$

$$\leq \left|\hat{p}_t(\bar{\boldsymbol{x}}_\kappa, \bar{f}) - p_t(\bar{\boldsymbol{x}}_\kappa, \bar{f})\right| + M\sum_{\bar{y}=1}^{L} \left|\left(\hat{p}_s(\bar{\boldsymbol{x}}_\kappa, \bar{f}, \bar{y}) - p_s(\bar{\boldsymbol{x}}_\kappa, \bar{f}, \bar{y})\right)\right|$$

Observe that, $\hat{p}_t(\bar{\boldsymbol{x}}_\kappa, \bar{f})$ is the standard empirical estimation of $p_t(\bar{\boldsymbol{x}}_\kappa, \bar{f})$. Therefore, applying Hoeffding's inequality, we have with probability $1 - \delta$,

$$|\hat{p}_t(\bar{\boldsymbol{x}}_\kappa, \bar{f}) - p_t(\bar{\boldsymbol{x}}_\kappa, \bar{f})| \leq \sqrt{\frac{\log 1/\delta}{2n_s}}$$

Similarly, with probability $1 - \delta$,

$$|\hat{p}_s(\bar{\boldsymbol{x}}_\kappa, \bar{f}, \bar{y}) - p_s(\bar{\boldsymbol{x}}_\kappa, \bar{f}, \bar{y})| \leq \sqrt{\frac{\log 1/\delta}{2n_t}}$$

Thus, with probability $1 - (\bar{v}^s L + \bar{v}^s L^2)\delta$, the above holds for any $\bar{f}, \bar{y}, \bar{\boldsymbol{x}}_\kappa$. Thus we have

$$|\hat{p}_t(\bar{\boldsymbol{x}}_\kappa, \bar{f}) - p_t(\bar{\boldsymbol{x}}_\kappa, \bar{f}) - \sum_{\bar{y}=1}^{L} w_J(\bar{\boldsymbol{x}}_J, \bar{y})(\hat{p}_s(\bar{\boldsymbol{x}}_\kappa, \bar{f}, \bar{y}) - p_s(\bar{\boldsymbol{x}}_\kappa, \bar{f}, \bar{y}))| \leq \sqrt{\frac{\log 1/\delta}{2n_s}} + LM\sqrt{\frac{\log 1/\delta}{2n_t}}$$

Combing this with inequalities B.1 and B.2, we have

$$\left(\hat{p}_t(\bar{\boldsymbol{x}}_\kappa, \bar{f}) - \sum_{\bar{y}=1}^{L} w_J(\bar{\boldsymbol{x}}_J, \bar{y})\hat{p}_s(\bar{\boldsymbol{x}}_\kappa, \bar{f}, \bar{y})\right)^2 - \left(p_t(\bar{\boldsymbol{x}}_\kappa, \bar{f}) - \sum_{\bar{y}=1}^{L} w_J(\bar{\boldsymbol{x}}_J, \bar{y})p_s(\bar{\boldsymbol{x}}_\kappa, \bar{f}, \bar{y})\right)^2$$

$$\leq 3LM\left(\sqrt{\frac{\log 1/\delta}{2n_s}} + LM\sqrt{\frac{\log 1/\delta}{2n_t}}\right)$$

Summing over $\kappa, \bar{\boldsymbol{x}}_\kappa, \bar{f}$, we have

$$\sum_{\kappa: J \subseteq \kappa \in [d], |K|=2s} \sum_{\bar{f}=1}^{L} \sum_{\bar{\boldsymbol{x}}_\kappa \in \mathcal{X}_\kappa} \left( \hat{p}_t(\bar{\boldsymbol{x}}_\kappa, \bar{f}) - \sum_{\bar{y}=1}^{L} w_J(\bar{\boldsymbol{x}}_J, \bar{y}) \hat{p}_s(\bar{\boldsymbol{x}}_\kappa, \bar{f}, \bar{y}) \right)^2$$

$$- \left( p_t(\bar{\boldsymbol{x}}_\kappa, \bar{f}) - \sum_{\bar{y}=1}^{L} w_J(\bar{\boldsymbol{x}}_J, \bar{y}) p_s(\bar{\boldsymbol{x}}_\kappa, \bar{f}, \bar{y}) \right)^2$$

$$\leq (2s)^s \bar{v}^s L \cdot 3LM \left( \sqrt{\frac{\log 1/\delta}{2n_s}} + LM \sqrt{\frac{\log 1/\delta}{2n_t}} \right)$$

That is to say, for all $w_J$, $|dd(J, w_J) - \hat{dd}(J, w_J)| \leq 3(2s\bar{v})^s ML^2 \left( \sqrt{\frac{\log 1/\delta}{2n_s}} + LM\sqrt{\frac{\log 1/\delta}{2n_t}} \right)$ with probability $1 - (\bar{v}^s L + \bar{v}^s L^2)\delta$. In addition, note that $dd(J, w_J)$ is a quadratic function over $w_J$. By the linear independence assumption, $dd(J, w_J)$ must be strongly convex. Now applying Lemma 4, we have

$$|dd(J, w_J^*) - \hat{dd}(J, \hat{w}_J^*)| \leq 3(2s\bar{v})^s ML^2 \left( \sqrt{\frac{\log 1/\delta}{2n_s}} + LM\sqrt{\frac{\log 1/\delta}{2n_t}} \right)$$

and

$$\|w_J^* - \hat{w}_J^*\|_2^2 \leq \frac{12}{\lambda}(2s\bar{v})^s ML^2 \left( \sqrt{\frac{\log 1/\delta}{2n_s}} + LM\sqrt{\frac{\log 1/\delta}{2n_t}} \right)$$

where $\lambda$ is the parameter corresponding to the strongly convexity of $dd(J, \cdot)$. This holds for a fixed $J$ with probability $1 - (\bar{v}^s L + \bar{v}^s L^2)\delta$. There are $\binom{d}{s}$ many possible choices of $J$. Thus, with probability at least $1 - d^s(\bar{v}^s L + \bar{v}^s L^2)\delta \geq 1 - 2d^s \bar{v}^s L^2 \delta$, the above holds. Replacing $2d^s \bar{v}^s L^2 \delta$ by $\delta$ gives the desired form. $\qquad\square$

Finally we are ready to prove the statement. By Lemma 5, $dd(I, w^*) = 0$ and for any $J \neq I$, we have $dd(J, w_J) > dd(I, w^*)$. Let $c_1 = \min_{J \neq I} d(J, w_J) > 0$, $c_2 = \frac{c_1}{6(2m\bar{v})^m ML^2}$, and the constant $c = c_2/c_1$. Now we make progresses in two steps.

- First let us show with high probability, the estimated shifted features match the true shifted features. This is equivalent to show, with high probability, $\hat{dd}(I, \hat{w}_I^*) < \hat{dd}(J, \hat{w}_J^*)$ for any $J \neq I$. To do so, let us note that, for any $J \neq I$,

$$\hat{dd}(I, \hat{w}_I^*) - \hat{dd}(J, \hat{w}_J^*)$$
$$= \hat{dd}(I, \hat{w}_I^*) - dd(I, w_I^*) - (\hat{dd}(J, \hat{w}_J^*) - dd(J, w_J^*)) + dd(I, w_I^*) - dd(J, w_J^*)$$
$$= \hat{dd}(I, \hat{w}_I^*) - dd(I, w_I^*) - (\hat{dd}(J, \hat{w}*_J) - dd(J, w_J^*)) + c_1$$

By Lemma 6, with probability $1 - \delta$, for any $J$,

$$|dd(J, w_J^*) - \hat{dd}(J, \hat{w}_J^*)|$$
$$\leq 3(2m\bar{v})^m ML^2 \sqrt{2m \log d + m \log \bar{v} + 2 \log L + \log 1/\delta} \left( \sqrt{\frac{1}{2n_s}} + LM\sqrt{\frac{1}{2n_t}} \right)$$
$$= \frac{1}{3} c_2 \sqrt{2m \log d + m \log \bar{v} + 2 \log L + \log 1/\delta} \left( \sqrt{\frac{1}{2n_s}} + LM\sqrt{\frac{1}{2n_t}} \right)$$

Therefore, we have

$$\hat{dd}(I, \hat{w}_I^*) - \hat{dd}(J, \hat{w}_J^*)$$
$$\leq -\frac{2}{3} c_2 \sqrt{2m \log d + m \log \bar{v} + 2 \log L + \log 1/\delta} \left( \sqrt{\frac{1}{2n_s}} + LM\sqrt{\frac{1}{2n_t}} \right) + c_1$$

By the assumption, $\sqrt{2m \log d + m \log \bar{v} + 2 \log L + \log 1/\delta} \left( \sqrt{\frac{1}{2n_s}} + LM \sqrt{\frac{1}{2n_t}} \right) < c_1/c_2$. Hence, we have

$$\hat{dd}(I, \hat{w}_I^*) - \hat{dd}(J, \hat{w}_J^*) \leq -\frac{2}{3} c_2 \frac{c_1}{c-2} + c_1 = \frac{1}{3} c_1 < 0$$

for any $J \neq I$. Thus, the correct shifted features are selected with probability at least $1 - \delta$.

- Finally we show the learned $\hat{w}_{\hat{J}}^*$ is close to the true importance weights $w^*$. By Lemma 6,

$$\|w_J - \hat{w}_J\|_2^2 \leq O \left( ML^2 \left( \sqrt{\frac{\log 1/\delta}{2n_s}} + LM \sqrt{\frac{\log 1/\delta}{2n_t}} \right) \right)$$

with high probability for all $J$. Thus it holds for the selected features $\hat{J}$. We have just shown that with high probability, the correct shifted features are selected, i.e., $\hat{J} = I$. Thus, it simply means

$$\|w^* - \hat{w}_{\hat{J}}^*\|_2^2 \leq O \left( ML^2 \left( \sqrt{\frac{\log 1/\delta}{2n_s}} + LM \sqrt{\frac{\log 1/\delta}{2n_t}} \right) \right)$$

which completes the proof.

$\square$

## C  Additional Discussions

Here we provide additional discussions.

**Motivating examples when SJS occurs.**   In Section 1 we give one example when SJS occurs. Now we give two more examples to show how SJS broadly exists in different scenarios.

- Cancer diagnosis: Suppose we wish to build an ML model to diagnose cancer based on patient health records. The model is developed based on labeled dataset in some developed countries. However, when deploying it to hospitals in a developing country, there might be much more young patients, and the cancer rate for the elderly can also increase. Suppose the other features' distribution remains unchanged given age and cancer diagnosis. Then the distribution shift is naturally an SJS.

- Toxic text recognition: Consider a mobile app that detects and filters toxic texts based on the content and senders' information. Due to unexpected events (for example, disappointing football games), the toxic texts rate, as well as the total number of texts, may both significantly increase in some locations at different time periods. The shift of text locations and toxic text rate is thus another example of SJS.

**Understanding sparse covariate shifts.**   Sparse covariate shift is a special case of covariate shift [31]. It occurs when the shifts are caused by a few variables. For example, consider two census datasets collected in two periods. If a large population moved from one city to another between the two periods and everything else remains the same, then there is a sparse covariate shift (location alone). It is also related to Adversarial patches: if adversarial noises are added to a few features (or a small number of pixels in image domains), it also corresponds to the sparse covariate shift.

## D  Experimental Details

Here we provide additional experimental details.

Table 2: Dataset statistics.

| Dataset | # of instances | # of features | Shift types |
|---------|---------------|---------------|-------------|
| BANKCHURN | 10000 | 10 | Synthetic |
| COVID-19 | 660787 | 8 | |
| CREDIT | 29946 | 23 | |
| EMPLOY | 227871 | 16 | Geography (CA,PR,IA,WI) |
| INCOME | 245783 | 10 | Geography (CO,CA,KS,OH) |
| INSURANCE | 32140 | 19 | Temporality (2014,2016,2018) |

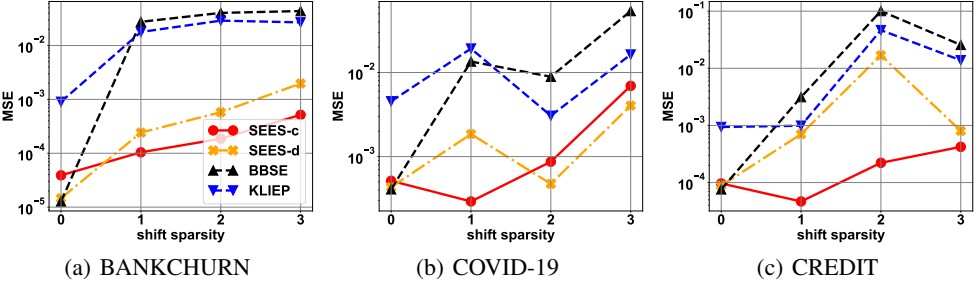

(a) BANKCHURN      (b) COVID-19      (c) CREDIT

Figure 5: Effects of shift sparsity. For each dataset, we measure how the $\ell_2$ loss of the estimated accuracy varies as number of shifted features increases given the same sample sizes. Overall, the estimation error of SEES slowly grows as shift sparsity increases, but is consistently lower than BBSE and KLIEP.

**Datasets and ML tasks.** We use six datasets for evaluation, namely, BANKCHURN [1], COVID-19 [2], and CREDIT [39] for various SJS simulations, and EMPLOY, INCOME, and INSURANCE [11] for performance evaluation under real world distribution shifts. BANKCHURN [1] contains 10 features such as gender, age, credit score and balance, and the goal is to predict whether a bank customer may churn. COVID-19 [2] is a subset of the publicly accessible COVID-19 dataset from the Israel government website, containing both demographic and symptom features. Here, we select the subset that contains all tested cases in January, 2022, and aim at predicting whether a person tests positive or negative for COVID-19. CREDIT [39] includes age, gender, education, bill payments and several other features for 29946 individuals. The goal is to predict the default payment. EMPLOY, INCOME, and INSURANCE are subset of the public use microdata samples from the US census [11]. EMPLOY contain 16 features for individual samples from four different states, CA, PR, IA, and WI and our goal is to predict if a person is employed or not. In INCOME, 245,783 anonymous census record samples from four states including CA, CO, KS, and OH are collected. The goal is to predict if a person's income is lower or higher than $50,000. INSURANCE contains 32140 individual samples from the state IA collected in year 2014, 2016, and 2018. The task is to predict whether an individual is covered by an insurance plan. The dataset statistics can be found in Table 2.

**Experiment setups.** All experiments were run on a machine with 20 Intel Xeon E5-2660 2.6 GHz cores, 160 GB RAM, and 80 GB disk with Ubuntu 18.04 LTS as the OS. Our prototype was implemented and tested in Python 3.8. To apply SEES-d, we discretized all continuous features. To apply SEES-c, we set the trade-off parameter $\eta = 0.001$. For continuous features, linear functions were used as the basis. For discrete features, indicate functions were adopted. For example, if $\bar{\boldsymbol{x}}_1 \in \mathbb{R}$ and $\bar{\boldsymbol{x}}_2 \in \{0, 1\}$, then the basis functions consist of three components, $\phi_1(\bar{\boldsymbol{x}}, y) = \bar{\boldsymbol{x}}_1, \phi_2(\bar{\boldsymbol{x}}, y) = \mathbb{1}_{\bar{\boldsymbol{x}}_2=0}$, and $\phi_3(\bar{\boldsymbol{x}}, y) = \mathbb{1}_{\bar{\boldsymbol{x}}_2=1}$. For KLIEP, the maximum number of iterations was set as 2,500.

**Effects of shift sparsity.** Figure 5 shows how the accuracy gap estimation performance vary as the number of shifted features increase. For each dataset, we start with shifting no features (label shift), to shifting 1, 2, and 3 features together with labels. Specifically, for BANKCHURN, shifts occur for (i) labels alone (0-SJS), (ii) then both labels and geography feature (1-SJS), (iii) then labels, geography, and gender (2-SJS), and (iv) finally labels, geography, gender, and credit card

owned before. For COVID-19, we start with shifting label alone (0-SJS), and then incrementally shift features age, gender, and contact risk to simulate 1-SJS, 2-SJS, and 3–SJS, respectively. For credits, 0-SJS, 1-SJS, 2-SJS, and 3-SJS correspond to shift in (i) labels, (ii) labels and marriage status, (iii) labels, marriage status, and gender, and (iv) labels, marriage status, gender, and credit balance. Overall, we observe that the estimation error of SEES-c and SEES-d slightly increases as the number of shifted features grows, but is consistently lower than that of BBSE and KLIEP.

**Robustness to data randomness.** In Section 5 we mainly focus on the average performance metric ($\ell_2$ loss). Here we provide additional robustness measurement: on the COVID-19 dataset, we repeat the case study experiments 200 times with different random seeds to generate the source and target datasets, and report the variance of the estimated accuracy gap. As shown in Table 3, we observe that the variance for all of the methods is small: variance of SEES-c and KLIEP are are less than 0.00003, and the variance for SEES-d is 0.003 and for BBSE is 0.0003.

Table 3: Variance of the estimated accuracy gap. The values were calculated over 200 experimental runs. Overall, the variance of all methods is small.

| Method | SEES-c | SEES-d | KLIEP | BBSE |
|---|---|---|---|---|
| Estimated Accuracy Gap Variance | 0.000018 | 0.0033 | 0.00005 | 0.0003 |

**Sensitivity of the sparsity parameter in SEES-d.** SEES-d needs knowledge of the sparsity parameter $s$, and thus a natural question is how sensitive its performance is when the sparsity parameter does not exactly match the true sparsity. To study this, we first generate a source-target pair (each containing 10,000 data points) on the COVID-19 dataset where labels and 3 features (age, contact risk, and gender) shift, and then measure the $\ell_2$ loss of the estimated performance gap by SEES-d with sparsity parameter ranging from 0 to 7 (the number of features). Here, sparsity parameter being 0, 1, 2, 4, 5, 6, and 7 corresponds to the mismatched case. Figure 6 summarizes the averaged $\ell_2$ loss over 100 experimental runs. The randomness comes from the choices of shifted features and the samples from source and target datasets, and the shaded area indicates the standard deviation. Overall, we observe that SEES-d is robust to small parameter mismatch: there is little change of the estimation error when the sparsity parameter (2, 3, 4, 5) is close to the true number of shifted features (3). When the parameter mismatch is too large, a relatively larger change in the estimation error can be observed (though SEES-d still works better than BBSE even when the model mismatch is large). This is because a too small sparsity parameter restricts the search space, while a too large parameter often incurs an identifiability issue as our theory shows (i.e., different feature-label joint distributions correspond to the same observed target feature distribution). In practice, if the user has a prior belief that the distribution shift is not sparse (i.e. number of shifted features is $> d/2$), then SEES-d may not be appropriate.

**Comparison with additional baselines.** For more in-depth understanding, we compare the performance of SEES with an additional baseline DLU [25]. DLU basically adopts discriminative learning on the union of the source and target dataset, and then uses the classifier's prediction to reweigh the source data. We adopt it on the COVID-19 dataset for a case study and measure the performance of the weight and accuracy gap estimation for it along with SEES-c, SEES-d, BBSE, and KLIEP. As shown in Table 4, DLU performs better than KLIEP, but is still much worse than SEES-c and SEES-d. For example, the MSE of its estimated weights is 0.320, while that of SEES-d is only 0.002. Hence, SEES-c and SEES-d still lead to the best estimated accuracy gap.

**Robustness across different shifts.** Real world distribution shifts may vary, and thus it is important to understand how different methods behave when encountering different shifts. To understand this, we study the performance of SEES-c and SEES-d along with all baselines (BBSE, KLIEP, and DLU) on the COVID-19 dataset when different shifts occur. Specifically, we generate source-target pair where (i) label shift, (ii) (sparse) covariate shift (feature age), and (iii) 1-SJS (both label and feature age) occurs, separately. Table 5 gives the squared $\ell_2$ loss of the accuracy gap estimation for all methods. Interestingly, we observe that SEES-c and SEES-d produce reliable accuracy gap estimation across different shifts, while all baselines are sensitive to shift types. For example, KLIEP and DLU achieves decent performance when there is only covariate shift, but their estimation is much worse when label shift occurs. Similarly, the estimation error of BBSE is small when labels

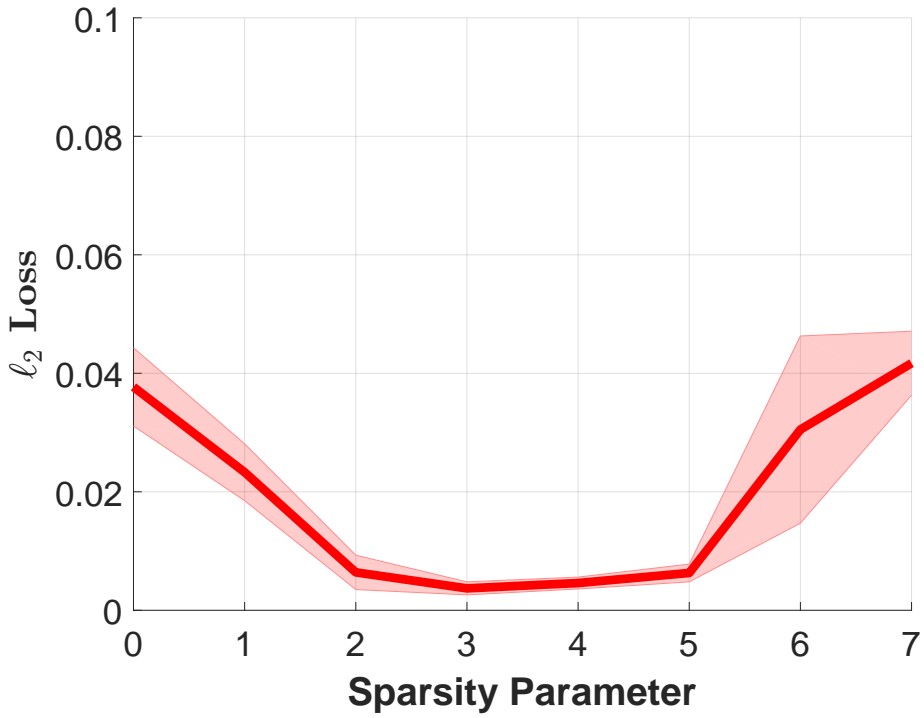

Figure 6: Sensitivity of the sparsity parameter in SEES-d on the COVID-19 dataset. On a source-target pair where labels and 3 features shift, we measure the $\ell_2$ loss of the estimated model performance gap produced by SEES-d with sparsity parameter ranging from 0 to 7 (number of features). The performance was averaged over 100 experimental runs, where the randomness comes from choices of shifted features and samples in the source-target pairs and the shaded area indicates the standard deviation.. Overall, SEES-d is robust to small parameter mismatch: there is little change of the estimation error when the sparsity parameter (2,3,4,5) is close to the true number of shifted features (3). In addition, the estimation performance of SEES-d with small parameter mismatch is also consistently better than all baselines.

Table 4: Performance of the weight and accuracy gap estimation on the COVID-19 case study. Overall, SEES-c and SEES-d achieve the smallest mean square error (MSE) and largest Pearson correlation coefficient (PCC). Thus, their estimated accuracy gap is closest to the true gap.

| Method | MSE | PCC | Est Gap | True Gap |
|--------|-----|-----|---------|----------|
| SEES-c | 0.003 | 0.996 | 18.4 | |
| SEES-d | *0.002* | *0.996* | *16.7* | |
| BBSE | 0.144 | 0.857 | 26.3 | 15.5 |
| KLIEP | 0.573 | -0.112 | 1.58 | |
| DLU | *0.320* | *0.328* | *0.38* | |

indeed shift but is much worse than other methods when the shift is due to covariate. On the other hand, the performance of SEES-c and SEES-d is as good as that of the best baseline when label or covariate shifts. For SJS, the proposed methods achieve significantly better estimation than all baselines. Those suggest that SEES-c and SEES-d are more robust to different shifts and thus safer to be deployed in the wild.

Table 5: Squared $\ell_2$ loss of the estimated accuracy gap on the COVID-19 dataset. SEES-c and SEES-d are the only approaches providing reliable estimation across all shifts.

| Method | Label Shift | Covariate Shift | Joint Shift |
|--------|-------------|-----------------|-------------|
| SEE-c  | 0.0005      | 0.0010          | 0.0003      |
| SEE-d  | 0.0004      | 0.0015          | 0.0019      |
| BBSE   | 0.0004      | 0.0321          | 0.0135      |
| KLIEP  | 0.0045      | 0.0008          | 0.0193      |
| DLU    | 0.0029      | 0.0001          | 0.0251      |

**More models on datasets with real world shifts.** Next we provide the performance estimation for more models on datasets with real world shifts. We study the estimation performance for three models, namely, a gradient boosting, a neutral network, and a decision tree. The maximum depth of the gradient boosting and deicision tree is 50, and the neutral network consists of two layers with 100 hidden units. As shown in Table 6, SEES often provides significant error reduction over the compared baselines BBSE and KLIEP.

Table 6: Root mean square error of estimated accuracy gap (%) under real shifts for gradient boosting, a neural network, and a decision tree. The numbers are averaged over all source-target pairs in each dataset. For each dataset and ML model, SEES provides significant estimation error reduction over baselines.

| Dataset | ML model | Accuracy estimation's $\ell_2$ error (%) | | | |
|---------|----------|--------|--------|------|-------|
| | | SEES-c | SEES-d | BBSE | KLIEP |
| EMPLOY | Gradient boosting | **2.9** | 3.0 | 5.2 | 5.2 |
| | Neural network | **5.2** | 6.0 | 6.2 | 5.6 |
| | Decision tree | 4.0 | **3.8** | 5.7 | 5.7 |
| INCOME | Gradient boosting | **1.9** | 2.4 | 3.0 | 3.4 |
| | Neural network | **4.4** | 6.4 | 9.8 | 7.6 |
| | Decision tree | **2.4** | 2.9 | 3.0 | 3.8 |
| INSURANCE | Gradient boosting | **1.7** | 2.2 | 2.1 | 5.0 |
| | Neural network | **2.2** | 4.7 | 10.8 | 7.3 |
| | Decision tree | **2.0** | 2.5 | 2.4 | 5.9 |