# OpenReview forum: "Estimating and Explaining Model Performance When Both Covariates and Labels Shift"
_NeurIPS.cc/2022/Conference — NeurIPS 2022 Accept_

### Official Review · Reviewer_9XVT · 2022-07-07

**Rating:** 5
**Confidence:** 4
**Soundness:** 2 fair
**Presentation:** 2 fair
**Contribution:** 3 good

**Summary:**

The performance of ML models may be degraded due to distribution shifts. To handle the covariate shift and label shift, prior work assumes the the type of the shit is known. In this paper, authors propose a joint way to measure covariate and label shift and explain model performance due to shifts without knowing wether its covariate or label shift. First, the paper describes when a joint shift is identifiable rigorously, then proposes an algorithmic framework to measure the shift in terms of importance weights (aka density ratio and likelihood ratio) under the assumption on the sparseness of the joint shift, and use the importance weights to estimate the model performance change. The proposed algorithm is evaluated based on six datasets along with two comparing methods, demonstrating the efficacy of the proposed algorithm. The claimed contributions include (1) proposing a new distribution shift model, i.e., a sparse joint shift (SJS), (2) proposing a general framework for performance shift estimation and explanation under SJS, and (3) demonstrating the efficacy of the proposed approach.

**Questions:**

* (minor) line 122: “s (|I| < s)” is confusing; s is an integer but looks like it is used as a function. I guess it’s better to write “s (i.e., |I| < s)”.

* Definition 1 is introduced in this paper; then real and motivational examples when SJS happens are needed. In particular, what is the real meaning of the invariance (i.e., the equation in line 122)? Is this invariance actually observable in practice?

* It would be better if the definition of the sparse covariate shift is clearly stated; it is partially appeared in line 153 (and it’s kind of understandable based on the context), but having a full definition would be better (e.g., p_s(x_I) \neq p_t(x_I) and p_s(y|x) = p_t(y|x). Moreover, what’s the real world example for the sparse covariate shift? I think this paper introduces many interesting concepts without motivational examples. It may be interesting if the sparse covariate shift can be motivated using adversarial patches.

* Is the identifiable (p_s, p_t) in Theorem 1 actually observable in practice?  It would be better if we can frequently observe the identifiable distributions in real datasets.

* The algorithm requires to know the sparsity of SJS, which cannot be known in advance. In this case, the sensitivity of the algorithm performance would be useful to justify the algorithm. Figure 5 embeds partial results on this, but it’s better to see when 0<=s<=d. If the algorithm is quite sensitive to s, what’s the rule of thumb on choosing s?

* The paper claims KLIEP [30] is the state of the art method for estimating the covariate shift; but KLIEP is a quite old paper (i.e, published in 2007), and it works under some assumptions (e.g., a basis (or kernel) function is properly chosen, which is always difficult). In this case, comparing other approaches is required. In particular, there are many papers on “Density Ratio Estimation” [R1], and especially a probabilistic classifier based density ratio estimation is quite simple and still used (e.g., [33] or [R2]). This could be a good baseline for the covariate shift experiments.


* I believe, to justify the efficacy of the proposed approach for SJS, three controlled experiments are required (at least): when the dataset contains (1) only covariate shift, (2) only label shift, and (3) both (sparse) covariate shift and label shift. For (1), the proposed approach needs to be as good as known density ratio estimators for covariate shift. For (2), the proposed approach needs to be as good as known approaches (e.g., [21]), But for (3), the proposed approach should be better than other shift dedicated approaches. Without these results, the proposed approach may not claim to be effective for both shifts when we don’t know the type of shifts. I believe synthetically generating datasets for (1-3) is not difficult (e.g., use exponential tilting in [33] for covariate shift).

[R1] Density Ratio Estimation in Machine Learning by Masashi Sugiyama and others.
[R2] https://arxiv.org/abs/2003.00343


**Limitations:**

I think the authors adequately addressed the limitations and potential negative societal impact of their work.

**Strengths And Weaknesses:**

### originality
I found that the attacking problem on estimating and explaining model performance under shifts is interesting and timely. Moreover, jointly considering the label and covariate shift under the umbrella of a sparse joint shift (SJS) looks like a novel attempt though the motivation on SJS is weak currently. The algorithm on estimating w(x,y) well exploits known techniques while adding a new component on the sparseness of SJS; but, comparison results are a bit weak. Measuring the performance shift by using the estimated w(x,y) is quite standard. See Questions.

### quality
The paper introduces a new shift, called SJS, which combines label and (sparse) covariate shifts, and proposes an algorithm for SJS, claiming that the proposed approach is good for handling both label and sparse covariate shifts. However, the empirical results do not support this claim well. I expect to see controlled experiments (e.g., experiments on a dataset with only covariate shift, a dataset with only label shift, and a dataset for both) to show the benefit of the proposed approach, but these results are missing. See Questions.

### clarity
The paper is overall well written, though I found a few points that would improve the clarity if fixed (e.g., the definition of sparse covariate shift, motivational examples on s-SJS, whether the identifiable SJS cases are practical, etc.); I’ll clarify more in the Questions section.

### significance
I think the crux of this paper is the introduction on SJS and its analysis on identifiability; it would be great if the paper includes clear justification that SJS can be frequently seen in practice and empirical supports that the proposed approach is useful for SJS.

---

> ### Author Response · Authors · 2022-08-02
> **Thank you for your thoughtful review and we answer your questions as follows. (1/2)**
>
> Thank you for your thoughtful review. We answer your questions as follows.
>
> ***[change “s (|I| < s)” to “s (i.e., |I| < s)” for clearness]***: Thanks for the suggestion. We have edited it accordingly and updated the draft.
>
> ***[Provide real and motivational examples when SJS happens. Explain the real meaning of the invariance.]***: We gave one motivating example in Section 1 (line 41). Other examples include:
>
> *Cancer diagnosis*: Suppose we wish to build an ML model to diagnose cancer based on patient health records. The model is developed based on a labeled dataset in some developed countries. However, when deploying it to hospitals in a developing country, there might be much more young patients, and the cancer rate for the elderly can also increase. Suppose the other features’ distribution remains unchanged given age and cancer diagnosis. Then the distribution shift is naturally an SJS.
>
> *Toxic text recognition*: Consider a mobile app that detects and filters toxic texts based on the content and senders’ information. Due to unexpected events (for example, disappointing football games), the toxic texts rate, as well as the total number of texts, may both significantly increase in some locations at different time periods. The shift of text locations and toxic text rate is thus another example of SJS.
>
> Those two examples have been added in the appendix, and a reference has been given after  Definition 1 (line 124). The invariance introduced in line 122 implies that only features in the set I and labels cause the distribution shift. One practical scenario is when the target dataset is a mixture of two datasets, where labels shift in the first and a few features shift in the second, compared to the source dataset. We also added more discussions in the appendix (see line 645- line 657, page 21).
>
> ***[Define the sparse covariate shift better. How is sparse covariate shift related to real-world examples and adversarial patches?]***: We added the formal definition of sparse covariate shift in line 152. Sparse covariate shift occurs when the shifts are caused by a few variables. For example, consider two census datasets collected in two periods. If a large population moved from one city to another between the two periods and everything else remains the same, then there is a sparse covariate shift (location alone). Adversarial patches are related: if adversarial noises are added to a few features (or a small number of pixels in image domains), it also corresponds to the sparse covariate shift. We added a detailed discussion in the appendix (see line 658-line 663, page 21).
>
> ***[Is the identifiable (p_s, p_t) in Theorem 1 actually observable in practice?]***: Yes. The identifiable condition basically requires conditional independence of non-shifted features, given the shifted ones. To verify this in practice, one can estimate the probability mass from the empirical dataset, and check if the matrix consisting of all corresponding probability mass is full rank.
>
> ***[How sensitive is the algorithm to sparsity s? If sensitive, how to choose s?]***: We provided a sensitivity analysis of SEES-d on the COVID-19 dataset in the appendix (see line 706-line 715, page 23). On a source-target pair where labels and 3 features shift, we evaluated the performance of SEES-d with sparsity parameter 3,2,1, the last two corresponding to model mismatches. Overall, the performance drops mildly. For example, setting the sparsity parameter to 2 only increases the error from 0.0026 to 0.0029. This result suggests that our algorithm is robust to some model mismatch. In general, we recommend picking the sparsity parameter to match the maximum number of shifted features derived from the applications.

---

> > ### Author Response · Authors · 2022-08-02
> > **Thank you for your thoughtful review and we answer your questions as follows. (2/2)**
> >
> > ***[Compare with additional baselines]***: We thank the reviewers for suggesting the additional baseline. We added the suggested baseline in [R2], based on discriminative learning on the union of source and target datasets (and we refer to it as DLU in the following). For the case study on the COVID-19 dataset, we found that DLU’s performance was similar to KLIEP and worse than SEES-d. We also measured DLU’s performance for the suggested controlled experiments. Details can be seen in our answer to the next question and the updated appendix (see line 715-line 740, Table 5 and Table 6, page 23- page 24).
> >
> >  [R2] Sangdon Park et al Calibrated Prediction with Covariate Shift via Unsupervised Domain Adaptation, 2020.
> >
> > ***[Compare all methods for covariate shifts, label shifts, and joint shifts]***: We measured the performance of SEES-c, SEES-d, BBSE, KLIEP, and the suggested baseline DLU [R2] on the dataset COVID-19 for all three different shifts. The results and detailed discussions can be found in the updated appendix (see line 723-line 738, page 23-page 24). Overall, our observations matched the reviewer’s expectation: for label shift, SEES-c and SEES-d matched BBSE’s performance and outperformed all other baselines, and for (sparse) covariate shift, the performance of SEES-c and SEES-d is close to KLIEP and DLU and is better than BBSE. When both covariates and labels shift, both BBSE-c and SEES-d significantly outperformed existing methods.
> >
> > *Thank you for your detailed feedback, which has enabled us to improve the paper. We would greatly appreciate it if you would consider increasing your score based on our detailed response. Please let us know if you have any further questions and we are happy to follow up.*

---

> > ### Comment · Reviewer_9XVT · 2022-08-05
> > **Thanks for the response**
> >
> > I appreciate the author's response. I think the response almost addresses my concerns, but there is a remaining concern on the algorithm's sensitivity over the sparsity parameter (which is also related to my main concern on the controlled experiment). Before the revision, I saw there is already the sensitivity analysis, but what I meant was that the result is not thorough to be convinced; in particular, I believe that COVD-19 dataset has at least 3 features, but Table 4 only considers three different sparsity parameters (i.e., 1,2, 3), assuming the true sparsity parameter is 3. As can be seen, when the sparsity parameter of the algorithm is 1, the result is quite worse than others. Letting the dimension of feature is d, if the true sparsity parameter is d/2, what is the sensitivity of SEED-d over the sparsity parameter of the algorithm from 0 to d?
> >
> > I believe that this sensitivity analysis is important as we never know the right sparsity parameter in advance. In this case, a user might simply choose the sparsity parameter of the algorithm s = d, which is equivalent to covariate shift thus it is not necessary to consider sparse-covariate shift.
> >
> >
> > ====== EDIT
> > Also, when the true sparsity parameter is d/2, d/2 features need to be randomly chosen.

---

> > > ### Author Response · Authors · 2022-08-09
> > > **Thank you for the prompt response! Additional results on robustness to sparsity mismatch.**
> > >
> > > We thank the reviewer for the prompt response and clarification. To further address the sensitivity question, we conducted additional experiments as the reviewer suggested: on the COVID-19 dataset, 3 randomly chosen features and the labels were shifted, and we evaluated SEES-d with the sparsity parameter ranging from 0 to 7 (the total number of features). Both source and target datasets contained 10,000 samples. Overall, we observe that SEEDS-d is robust to small parameter mismatch: there is little change in the estimation error when the sparsity parameter (2, 3, 4, 5) is close to the true number of shifted features (3).
> > >
> > > When the parameter mismatch is too large, a relatively larger change in the estimation error can be observed (though SEES-d still works better than BBSE here). This is because a too small sparsity parameter restricts the search space, while a too large parameter often incurs an identifiability issue as our theory shows (i.e., different feature-label joint distributions correspond to the same observed target feature distribution). We provide more details in the Appendix (see line 706 - line 723 and Figure 6, page 23-page 24).
> > >
> > > In general, all methods need some assumptions on the distribution shifts since the general data distribution shifts are not identifiable (as discussed on line 108 - line 112, in Section 3). Previous approaches are actually more sensitive to the sparsity parameter due to their stronger assumptions (e.g., BBSE assumes label shift and no feature shifts are allowed). In practice, users can empirically verify whether the sparsity hyperparameter in SEES is close to the true sparsity by comparing the target distribution with the source distribution adjusted by the selected shift features. We have added discussions of this to the text.
> > >
> > > Thank you again for your feedback, we really appreciate it! Please let us know if you have further questions and we are happy to follow up.

---

> > > > ### Comment · Reviewer_9XVT · 2022-08-09
> > > > **Thanks!**
> > > >
> > > > Thanks for the additional sensitivity analysis. I have increased my score based on the current paper status and the following reasons.
> > > >
> > > > * Based on Figure 6 of Appendix, the SEED-d is robust if there is small parameter mismatch. I believe that the proposed algorithm can be further improved by increasing the searching space (i.e., |J| <= m instead of |J|=m); this significantly increases the computational time, but sampling may help. Given this computationally expensive algorithm, we can say that a user needs not to consider the sparsity parameter too much, assuming d/2 features are at most shifted (as also mentioned in line 722 of Appendix). In short, the algorithm needs to be improved more to be robust to the choice of the sparsity parameter, but I believe that it is not too difficult. (BTW, it is better to highlight the limitations of the algorithm in the main paper)
> > > > * The controlled experiment in Table 5 considers the best case of the proposed algorithm (e.g., when the true sparsity is known). It’s better to have the same experiment when the true sparsity is unknown (BTW, DLU is well-known at least in [R1]; [23] may not be a right citation unless there is a reason).
> > > > * Assuming that the paper demonstrates the possibility of detecting SJS by proposing a few (but possibly naive) algorithms (i.e., proof by demonstration), I think the limitation on the current algorithm is okay. But, the paper may need to be written in this way.
> > > > * Finally, but most importantly, I think that the new concept SJS (and related theories) could contribute to the community to detect shifts better under the sparsity assumption, which compensates for the algorithmic limitations.
> > > >
> > > > Thanks for the contributions!

---

> ### Author Response · Authors · 2022-08-05
> **We would like to hear back from reviewer 9XVT**
>
> Dear reviewer 9XVT
> We would like to follow up to see if our response addresses your concerns or if you have any further questions. We would really appreciate the opportunity to discuss this further if our response has not already addressed your concerns. Thank you again!

---

### Official Review · Reviewer_erZL · 2022-07-08

**Rating:** 7
**Confidence:** 4
**Soundness:** 4 excellent
**Presentation:** 3 good
**Contribution:** 4 excellent

**Summary:**

This paper introduces Sparse Joint Shift (SJS), a new distribution shift model that detects both label and covariate shifts simultaneously. The authors show how the proposed framework includes existing distribution shifts models and discuss under what assumptions SJS is identifiable. Overall, this paper is very novel and generalize the existing distribution shift approaches in an elegant way.  The authors also introduce a new algorithm able to detect SJS which performs better than the existing distribution shift models. The paper has strong theoretical and empirical discussion and I enjoyed reading it very much.

**Questions:**

To prevent confusion with the source s, I suggest authors use a different letter to denote the size of the set.

**Strengths And Weaknesses:**

Strengths: 1- Novelty, 2- Theoretical foundation, 3- experiments.

Weakness: 1- If space allowed, it would have been refreshing to see experiments on more complex data sets.

---

> ### Author Response · Authors · 2022-08-02
> **Thank you for your strong support of our paper, and we are happy that you enjoyed reading it!**
>
> Thank you for your strong support of our paper, and we are happy that you enjoyed reading it!
>
> ***[use a different letter to denote the size of the set]***: Thank you for this suggestion. Yes, we have changed the notation to m.

---

### Official Review · Reviewer_wUbV · 2022-07-12

**Rating:** 6
**Confidence:** 4
**Soundness:** 3 good
**Presentation:** 3 good
**Contribution:** 3 good

**Summary:**

The authors propose an interesting model of data shift which only label and a small set of features change between the training and test distributions. They design simple distribution matching algorithms enforcing shift sparsity and show that they work well in sparse shift scenarios.



**Questions:**

- Using linear features in SEES-c appear odd as x_k can be negative, making the weight function w(x,y) negative. Should there be extra assumptions?
- The authors should provide some evaluations when the sparse-shift assumption is not entirely correct, as the sparse shift assumption can be difficult/expensive to verify in practice. For example, what would happen if we assume a shift of 2 features only when there are 4 shifted features? Would the performance of the method degrade gracefully?


**Limitations:**

- The main limitation of the method is knowing when the sparse-shift assumption is true so that the method can be applied. The discrete version SEES-d can also grow computationally expensive as the size of the shifted features s grows.


==========

After revision, I am keeping my recommendation of weak accept. The authors have addressed most of the questions raised by the reviewers and performed additional experiments.


**Strengths And Weaknesses:**

- The proposed model of shift is realistic and covers target shift and a part of covariate shift.
- The proposed algorithms SEES-c and SEES-d are natural for continuous and discrete features, and manage to discover the underlying shifts in the scenarios evaluated in the experiments.
- For weakness, SEES-d does not scale well with size of shifted feature set s, as it requires searching over all subsets J of size at most s.
- The authors evaluate mostly on artificially created shift scenarios when they know the ground truth has only a few features shifted. It is not clear how the algorithm would perform in more complicated scenarios, such as medical data collected from different hospitals.

---

> ### Author Response · Authors · 2022-08-02
> **Thank you for your helpful feedback and support for the paper!**
>
> Thank you for your helpful feedback and support for the paper! We answer your questions below and we have updated the paper to incorporate your suggestions.
>
> ***[Using linear features in SEES-c appear odd as x_k can be negative, making the weight function w(x,y) negative. Should there be extra assumptions?]***: x_k needs to be non-negative, which holds in our empirical studies (e.g., age/salary is always non-negative). In general, we can transform negative x_k to non-negative values, e.g., via affine transformation for finite data. We clarified this point in the paper (see line 201, page 6).
>
> ***[The authors should provide some evaluations when the sparse-shift assumption is not entirely correct]***: We evaluated the sparsity robustness of SEES-d on the COVID-19 dataset. Specifically, on a source-target pair where the labels and 3 features all shifted, we measured the accuracy estimation error of SEES-d with sparsity parameter being 3, 2, and 1, where the last two correspond to model mismatches. Overall, the performance drops mildly as the sparsity parameter decreases.  For example, SEES-d with the sparsity parameter being 2 only increases the l-2 error from 0.0026 (when the parameter matches the true sparsity) to 0.0029. This suggests that the method is relatively robust to model mismatch. The details can be found in the Appendix (see line 706-line 723, page 23).

---

### Official Review · Reviewer_8kzf · 2022-07-13

**Rating:** 6
**Confidence:** 2
**Soundness:** 3 good
**Presentation:** 4 excellent
**Contribution:** 3 good

**Summary:**

This paper deals with the interesting and important problem of distribution shift. The authors formalize a model (Sparse Joint Shift, SJS) which unifies existing shift models by considering joint shift of both labels and a few features.The authors show conditions for identifiability of models under these shifts, and propose Shift Estimation and Explanation under SJS (SEES), a method to estimate the empirical performance gap and explain it in terms of shifted features. The authors show empirical benefits of SEES compared to BBSE and KLIEP baselines.

**Questions:**

- Can you expand the empirical results with robustness measures (e.g. variance over multiple experimental runs)?

- How should we consider extensions of this framework (for instance, when we have access to multiple dataset shifts and possible task representations)?

**Limitations:**

This paper is appropriately presented in the context of related work. Limitations of the SJS framework are briefly discussed in the Conclusion but could be discussed more straightforwardly throughout the paper.



===Edit after revision===

After reading the other reviews and responses, I would keep my recommendation of acceptance.

Thank you for reporting the variance in experimental results. These suggest very stable training.

I agree with reviewer 9XVT about comparing the methods across many types of shift and thank the authors for including some of this comparison in the updated appendix. I believe this is an important topic. I am also interested in reviewer 9XVT's question about the sensitivity to sparsity parameter s and it now seems to me that this may limit the model to be used in applications where we can evaluate performance wrt this hyperparameter (eg by cross-validation).

**Strengths And Weaknesses:**

Strengths:

The paper is written well, with straightforward assumptions and conclusions, and was enjoyable to read. The problem addressed is important and the provided SJS framework appears to be a useful way to approach this problem. I also appreciate the identifiability condition and the discussion of implications.


Weaknesses:

The empirical measurements are rather shallow, although measured over a few different datasets. All empirical measurements should have robustness measures, but are currently presented as only point estimates. Finally, when claiming seeking as a desired characteristic of the method, it would be good to evaluate the explainability.

---

> ### Author Response · Authors · 2022-08-02
> **Thank you for your helpful feedback and support of the paper!**
>
> Thank you for your helpful feedback and support for the paper! We answer your questions below and we have updated the paper to incorporate your suggestions.
>
>
>
> ***[Can you expand the empirical results with robustness measures (e.g. variance over multiple experimental runs)?]***: Yes. We measured the variance of the performance estimation error over 200 experimental runs with 10,000 samples on the COVID-19 dataset. Overall, we observe that the variance for all of the methods is small: the variance of SEES-c and KLIEP is less than 0.00003 and the variance for SEES-d is 0.003 and for BBSE is 0.0003.  The variance is relatively large for SEES-d and BBSE.  We have added this information and more details in the updated appendix (see line 700-line 706, page 23).
>
> ***[How should we consider extensions of this framework (for instance, when we have access to multiple dataset shifts and possible task representations)?]***: Our framework is extendable to other scenarios and opens many interesting follow-up research questions. For multiple dataset shifts, for example, it would be interesting to augment the identification of shifted covariates and labels by correlations between different datasets.  The proposed framework may also help explain how different task representations relate to each other.

---

> ### Author Response · Authors · 2022-08-09
> **Thank you for the feedback! Additional experiments showing robustness to sparsity parameter.**
>
> Thank you for the feedback after our initial response! We conducted additional sensitivity experiments as suggested by reviewer 9XVT. Overall, we find that SEES-d is robust to mismatch in the sparsity parameter. Across 100 experiments, there is relatively little change in the estimation error when the sparsity hyperparameter is 2, 3, 4, or 5 when the true number of shifted features is 3. More details can be found in the Appendix (see line 706 - line 723 and Figure 6, page 23-page 24).
>
> In general, all methods need some assumptions on the distribution shifts since the general data distribution shifts are not identifiable (as discussed on line 108 - line 112, in Section 3). Previous approaches are actually more sensitive to the sparsity parameter due to their stronger assumptions (e.g., BBSE assumes label shift and no feature shifts are allowed). In practice, users can empirically verify whether the sparsity hyperparameter in SEES is close to the true sparsity by comparing the target distribution with the source distribution adjusted by the selected shift features. We have added discussions of this to the text.
>
> Thank you again for your feedback, we really appreciate it! Please let us know if you have further questions and we are happy to follow up.

---

### Meta-Review · Area_Chair_rida · 2022-08-26

**Recommendation:** Accept
**Confidence:** Less certain

**Metareview:**

The authors study the important problem of distribution shift under a new SJS model. Identifiability results are proved and empirical experiments illustrate the value of the proposed model. During discussion, some concerns on the experiments were addressed. Overall, there was a weak consensus to accept this paper, which I concur with.

**Award:**

No

---

### Decision · Program_Chairs · 2022-09-14

Accept